# A mechanism for exocytotic arrest by the Complexin C-terminus

Mazen Makke[1†], Maria Mantero Martinez[1], Surya Gaya[1], Yvonne Schwarz[1], Walentina Frisch[1], Lina Silva-Bermudez[1], Martin Jung[2], Ralf Mohrmann[3], Madhurima Dhara[1†*], Dieter Bruns[1†*]

[1]Institute for Physiology, Center of Integrative Physiology and Molecular Medicine, University of Saarland, Homburg, Germany; [2]Institute for Medical Biochemistry and Molecular Biology, University of Saarland, Homburg, Germany; [3]Institute for Physiology, Otto-von-Guericke University, Magdeburg, Germany

**Abstract** ComplexinII (CpxII) inhibits non-synchronized vesicle fusion, but the underlying mechanisms have remained unclear. Here, we provide evidence that the far C-terminal domain (CTD) of CpxII interferes with SNARE assembly, thereby arresting tonic exocytosis. Acute infusion of a CTD-derived peptide into mouse chromaffin cells enhances synchronous release by diminishing premature vesicle fusion like full-length CpxII, indicating a direct, inhibitory function of the CTD that sets the magnitude of the primed vesicle pool. We describe a high degree of structural similarity between the CpxII CTD and the SNAP25-SN1 domain (C-terminal half) and show that the CTD peptide lowers the rate of SDS-resistant SNARE complex formation in vitro. Moreover, corresponding CpxII:SNAP25 chimeras do restore complexin's function and even 'superclamp' tonic secretion. Collectively, these results support a so far unrecognized clamping mechanism wherein the CpxII C-terminus hinders spontaneous SNARE complex assembly, enabling the build-up of a release-ready pool of vesicles for synchronized $Ca^{2+}$-triggered exocytosis.
DOI: https://doi.org/10.7554/eLife.38981.001

*For correspondence:
madhurima.dhara@uks.eu (MD);
dieter.bruns@uks.eu (DB)

†These authors contributed equally to this work

Competing interests: The authors declare that no competing interests exist.

## Introduction

The accumulation of vesicles in a release-ready state is essential for fast transmitter release from secretory cells. Yet, the molecular mechanisms of how the assembly of membrane-bridging and fusion promoting SNARE complexes is paused in a coordinated fashion to allow for fast synchronous release upon intracellular $Ca^{2+}$-elevations have remained unclear. Complexin (Cpx) is a small cytosolic α-helical protein that binds to assembled SNARE complexes with high affinity and has multiple domains with controversially discussed functions (*Brose, 2008*; *Mohrmann et al., 2015*; *Zhou et al., 2017*). The central α-helix of Cpx binds to the groove between the helices of Syntaxin I (SyxI) and Synaptobrevin II (SybII) within the SNARE complex (*Bracher et al., 2002*; *Chen et al., 2002*) and is essential for facilitatory as well as inhibitory effects of Cpx action in neurotransmission. The N-terminus of Cpx accelerates evoked release in murine neurons and neuroendocrine cells (*Dhara et al., 2014*; *Maximov et al., 2009*; *Xue et al., 2007*) by increasing the $Ca^{2+}$-affinity of synchronous secretion, but has no effect at the neuromuscular junction (NMJ) of C. elegans (*Hobson et al., 2011*; *Martin et al., 2011*). The accessory α-helix, instead, has been shown to play an inhibitory action in in vivo studies (*Cho et al., 2014*; *Martin et al., 2011*; *Maximov et al., 2009*; *Trimbuch et al., 2014*; *Xue et al., 2007*; *Yang et al., 2010*). A variety of different models for the inhibition by the accessory α-helix have been proposed including direct binding to SNAREs or other proteins (*Bykhovskaia et al., 2013*; *Cho et al., 2014*; *Giraudo et al., 2009*; *Krishnakumar et al., 2011*; *Kümmel et al., 2011*; *Lu et al., 2010*; *Yang et al., 2010*), electrostatic membrane interactions

(*Trimbuch et al., 2014*) or by stabilizing the secondary structure of the central helix (*Radoff et al., 2014*).

The C-terminal region of Cpx, which comprises almost half of the protein, binds to synaptotagminI (SytI; [*Tokumaru et al., 2008*]) and to phospholipids (*Malsam et al., 2009*; *Seiler et al., 2009*; *Snead et al., 2014*; *Wragg et al., 2013*). It has been shown to clamp spontaneous fusion in neurons (*Cho et al., 2010*; *Kaeser-Woo et al., 2012*; *Martin et al., 2011*) and to hinder premature secretion in neuroendocrine cells (*Dhara et al., 2014*). In particular, the amphipathic helix motif within Cpx's C-terminus has been suggested to position Cpx to synaptic vesicles in a curvature sensitive manner, thereby concentrating other inhibitory domains of Cpx (e.g. accessory α helix) at the fusion site (*Gong et al., 2016*; *Snead et al., 2014*; *Wragg et al., 2013*). While this form of proximity accelerated inhibition appears attractive to catalyze a reliable blockade of SNARE assembly, recent structure function analyses with C. elegans Cpx revealed that membrane binding is important but not sufficient for Cpx inhibitory effects (*Snead et al., 2017*; *Wragg et al., 2017*), leaving room for unknown interactions of the Cpx C-terminus that are instrumental for arresting vesicle fusion.

Here, we set out to delineate the Cpx C-terminus action in vesicular release of mouse chromaffin cells. Using viral expression of a truncated Cpx-variant (lacking the far C-terminus) or acute infusion of an isolated C-terminal peptide (aa101-134) in wt chromaffin cells, we show that the C-terminal domain (CTD) of CpxII is essential and rate-limiting for hindering premature fusion and for augmenting a pool of primed vesicles. Furthermore, the clamping function of CpxII can be reconstituted by N-terminal domains of CpxII and its far CTD as separate fragments in CpxII-ko cells. Thus, physical continuity through the length of CpxII is not required and composite CpxII domains can act in tandem to establish a fully functional ensemble. We point out that the CpxII CTD exhibits a high degree of structural similarity to the C-terminal half of the SNAP25-SN1 domain and show that it lowers the rate of SNARE complex formation in vitro. Moreover, CpxII:SNAP25-SN1 (C-terminal half) chimeras fully restore function in CpxII deficient cells. Collectively, these results provide evidence for a new model wherein the CpxII C-terminus competes with SNAP25-SN1 for binding to the SNARE complex and thereby halts progressive SNARE complex formation before the triggering $Ca^{2+}$-stimulus.

## Results

### The C-terminus of CpxII (amino acid 101–134) is rate-limiting for inhibition of vesicle fusion

To probe the function of CpxII's C-terminus in fast $Ca^{2+}$-dependent exocytosis, we recorded membrane capacitance (CM) increase in response to photolytic $Ca^{2+}$-uncaging (NP-EGTA) in cultured mouse chromaffin cells. Flash-induced changes in $[Ca^{2+}]i$ were monitored with a combination of calcium indicators (Fura-2 and Furaptra). Our previous results have implicated the CpxII C-terminus (amino acid 73–134) in suppression of tonic release (*Dhara et al., 2014*). To narrow down the region responsible for inhibitory function of CpxII, we generated a truncated variant (Cpx$^{1-100}$) lacking the last 34 amino acids of the protein, which contains a glutamate cluster (putative SytI interaction site, [*Tokumaru et al., 2008*]) and an amphipathic helical region (*Snead et al., 2014*) (*Figure 1A,B*). Viral expression of full-length CpxII in CpxII knock out (CpxII ko) cells reconstituted synchronous release and induced a pronounced exocytotic burst (EB) comprising two kinetically distinct components - the readily releasable pool [RRP] and the slowly releasable pool [SRP] (*Figure 1C,D*). In contrast, the Cpx$^{1-100}$ mutant largely failed to restore either component of the EB when compared to CpxII ko responses (*Figure 1C,D*). However, the slow kinetic rates of RRP and SRP (determined from the detailed fitting of the CM responses) and longer exocytotic delay seen with CpxII ko cells were fully reversed with expression of Cpx$^{1-100}$ mutant (*Figure 1E*), confirming our previous observation that the N-terminal domains of CpxII are responsible for accelerating the kinetics of synchronous secretion (*Dhara et al., 2014*). Furthermore, Cpx$^{1-100}$ failed to suppress tonic secretion at submicromolar $[Ca^{2+}]i$ as observed in CpxII ko cells (*Figure 1F,G*). Thus, the last 34 amino acids at the C-terminal end of CpxII are instrumental in arresting premature fusion at submicromolar $[Ca^{2+}]i$, thereby establishing a pool of release ready vesicles that is rapidly secreted upon the $Ca^{2+}$-trigger. Expression of CpxII in wt cells significantly diminished premature secretion and strongly boosted the EB component upon $Ca^{2+}$-uncaging (*Figure 1—figure supplement 1*). In contrast, expression of Cpx$^{1-100}$ in wt cells enhanced tonic release and consequently decreased the subsequent synchronous secretion

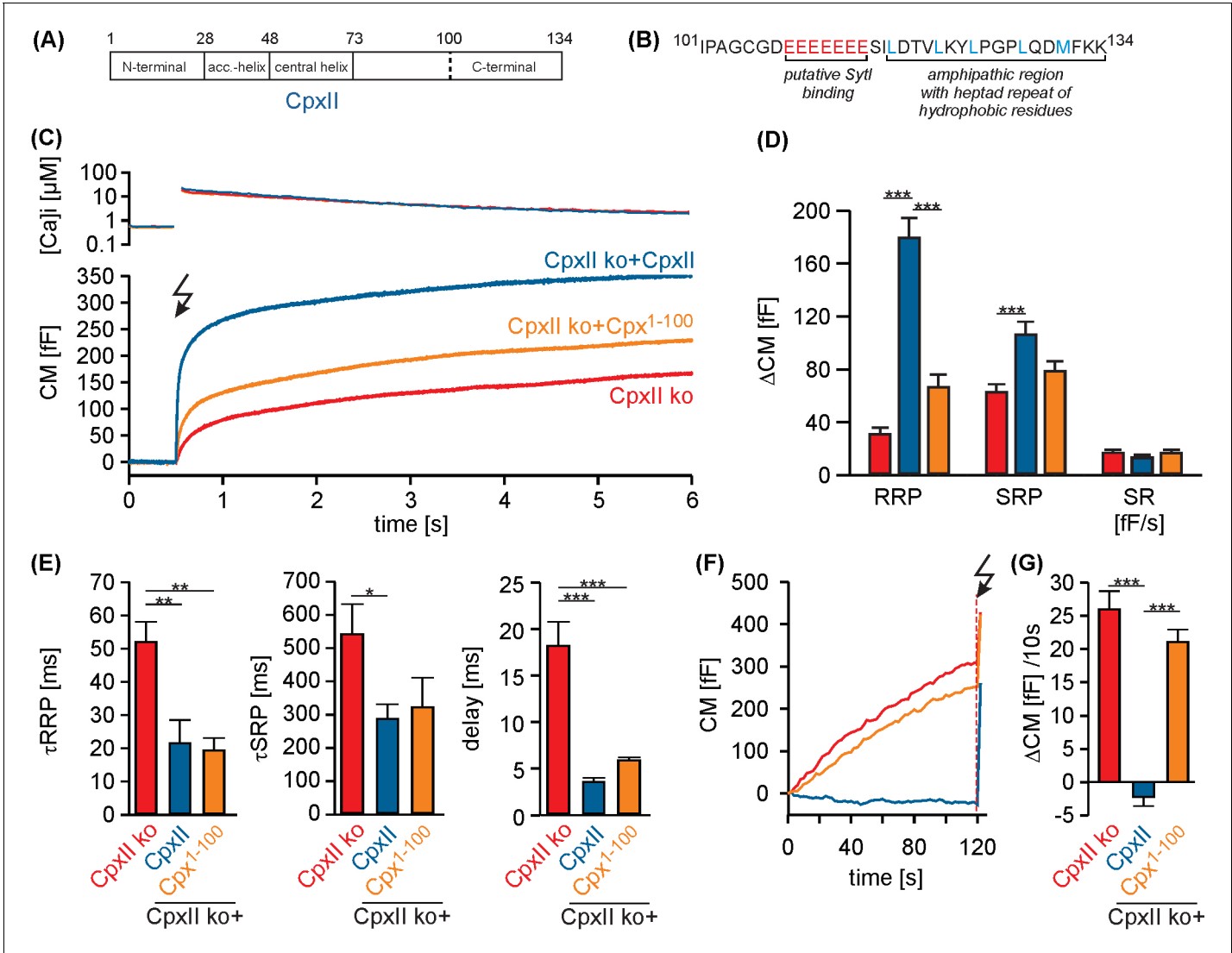

**Figure 1.** The C-terminal domain of CpxII controls the magnitude of synchronous secretion. (**A**) Schematic view on CpxII subdomains (numbers indicate amino acid positions within CpxII). (**B**) Primary sequence of the CpxII C-terminal region (residue 101–134) highlighting its structural characteristics (glutamate cluster, red; heptad repeat of hydrophobic residues, blue). (**C**) Mean $[Ca^{2+}]i$ levels (top) and corresponding CM signals (bottom) of CpxII ko cells (n = 25) and those expressing CpxII (n = 32) or the mutant $Cpx^{1-100}$ (n = 29). Flash is at t = 0.5 s (arrow). (**D**) Amplitudes of RRP and SRP, the rate of sustained release (SR; femtofarad/second) determined for CpxII ko (red), CpxII ko + CpxII (blue), CpxII ko+$Cpx^{1-100}$ (orange). (**E**) The time constants for the EB components ($\tau$RRP and $\tau$SRP), and the exocytotic delay. (**F**) Mean tonic CM traces of the cells shown in (**C**) before the triggering flash response (arrow). (**G**) The rate of tonic exocytosis (determined at similar $[Ca^{2+}]i$: in nM, CpxII ko: 714 ± 32; CpxII ko + CpxII: 639 ± 23; CpxII ko+ $CpxII^{1-100}$: 628 ± 30) is significantly reduced with CpxII but not with its mutant. ANOVA followed by Tukey-Kramer post-hoc test. *p<0.05; **p<0.01; ***p<0.001. Error bars indicate mean ± SEM.

DOI: https://doi.org/10.7554/eLife.38981.002

The following figure supplements are available for figure 1:

**Figure supplement 1.** $CpxII^{1-100}$ competes with the endogenous CpxII for the control of secretion.

DOI: https://doi.org/10.7554/eLife.38981.003

**Figure supplement 2.** CpxII CTD clamps tonic release and hinders the expansion of the initial fusion pore.

DOI: https://doi.org/10.7554/eLife.38981.004

response (*Figure 1—figure supplement 1*). This illustrates a dominant negative effect of the $Cpx^{1-100}$ mutant, which binds to productive SNARE complexes and out-competes the endogenous protein for control of tonic secretion. To substantiate these findings, we used simultaneous CM and carbon fiber amperometry recordings that allow for independent measurements of secretion from the same

cell. Again, expression of CpxII in wt cells suppressed tonic exocytosis, whereas Cpx[1-100] mutant increased both CM responses and amperometric event frequency beyond the level of wt cells (*Figure 1—figure supplement 2A,B*). These results demonstrate that the CpxII C-terminus is able to 'clamp' high rates of tonic vesicle fusion. The close correlation between ΔCM and event frequency (slope: wt 0.17 events/fF, $r^2$ 0.82; wt + CpxII 0.16, $r^2$ 0.79; wt + Cpx[1-100] 0.16, $r^2$ 0.77) further showed that the observed CM changes were due to exocytosis of catecholamine-containing granules. Thus, expression of neither wt nor the mutant Cpx variant affected the mode of exocytosis (kiss and run vs full fusion).

Detailed analyses of single amperometric events report distinct kinetic phases of cargo release from individual chromaffin granules, consisting of an initial slow release phase through a nascent pore (prespike) followed by a phase of rapid release reflecting bulk catecholamine discharge through a widened pore (main spike) (*Bruns and Jahn, 1995*; *Chow et al., 1992*). In good agreement with our previous findings (*Dhara et al., 2014*), expression of CpxII in wt cells prolonged the prespike duration, and increased prespike charge (*Figure 1—figure supplement 2D–F*), but left the main spike properties unaltered (*Figure 1—figure supplement 2C*). Prespike signals are characterized by fast fluctuations in the current trace that clearly exceed the baseline noise and report transient changes in neurotransmitter flux through the early fusion pore (*Kesavan et al., 2007*). These fluctuations likely reflect unsuccessful attempts of the SNARE machinery to widen the initial fusion pore as they are susceptible to diminished force transfer by the SNARE proteins onto the merging membranes (*Dhara et al., 2016*; *Kesavan et al., 2007*). Expression of CpxII similarly suppressed the current fluctuations and reduced the rms noise (a threshold independent parameter) during the prespike signal (*Figure 1—figure supplement 2E*), consistent with the view that CpxII directly acts on SNAREs and suggesting that its inhibitory action is continued even after the initiation of fusion.

Unlike full-length CpxII, the CpxII[1-100] mutant protein failed to alter fusion pore dynamics. Neither the rate of fusion pore expansion nor its current fluctuations were significantly affected when compared with control (*Figure 1—figure supplement 2F–H*), indicating that the CpxII CTD actively controls both the magnitude of tonic secretion, as well as the kinetics of fusion pore expansion.

## Acute inhibitory effects of the CpxII C-terminus

Given the strong phenotype of the truncation mutant (Cpx[1-100]), we next generated oligopeptides, representing the CpxII CTD (amino acids 101–134, CTD-peptide) and a control peptide with a scrambled sequence (scr-peptide, see Materials and methods). Compared with the scr-peptide, acute infusion of the CTD-peptide (10 µM) via the patch pipette into wt cells enhanced both, RRP and SRP components of the EB without changing kinetics of release in $Ca^{2+}$-uncaging experiments (*Figure 2A–C*). Furthermore, the CTD-peptide arrested tonic exocytosis at submicromolar $[Ca^{2+}]i$ (*Figure 2D,E*) and also inhibited high rates of tonic release seen with 19 µM $[Ca^{2+}]i$ (*Figure 2E,F*). Moreover, like viral expression of full-length CpxII, infusion of the CTD-peptide in wt cells also lengthened the prespike duration as depicted for exemplary amperometric events (*Figure 2G*) compared to the scr-peptide. Cell-weighted averages and cumulative frequency distributions confirmed that CTD-peptide increases prespike charge and dampens fusion pore fluctuations, without altering the properties of main amperometric spikes (*Figure 2H*, *Figure 2—figure supplement 1*). Of note, in a subset of recordings the initial CM increase of CTD peptide infused cells was similar to that of controls, before it rapidly declined to a slower, sustained increase in CM (*Figure 2E*, inset). Based on an estimated time constant of around 60 s for peptide infusion into chromaffin cells ($R_{access}$11.8 ± 0.2 MΩ; CM 4.0 ± 0.1 pF; [*Pusch and Neher, 1988*]), this behavior suggests that saturating concentrations for inhibitory peptide action are readily reached after establishment of whole-cell configuration. Since in vitro single vesicle-vesicle fusion assays have shown that Cpx suppresses spontaneous fusion at concentrations as low as 0.5 µM (*Lai et al., 2014*), it is likely that the pipette concentration of CTD-peptide (10 µM) used in our experiments exceeds the required effective concentration of CpxII-CTD for inhibiting vesicle fusion. Collectively, these results show that the isolated CpxII CTD mimics the phenotype observed with CpxII overexpression in wt cells (compare with *Figure 1—figure supplement 1* and *Figure 1—figure supplement 2*). They suggest that the CpxII CTD (despite the presence of endogenous CpxII) is a rate limiting factor for suppression of premature release (thereby defining the magnitude of $Ca^{2+}$-triggered synchronous exocytosis) and decelerates neurotransmitter discharge from fusing vesicles.

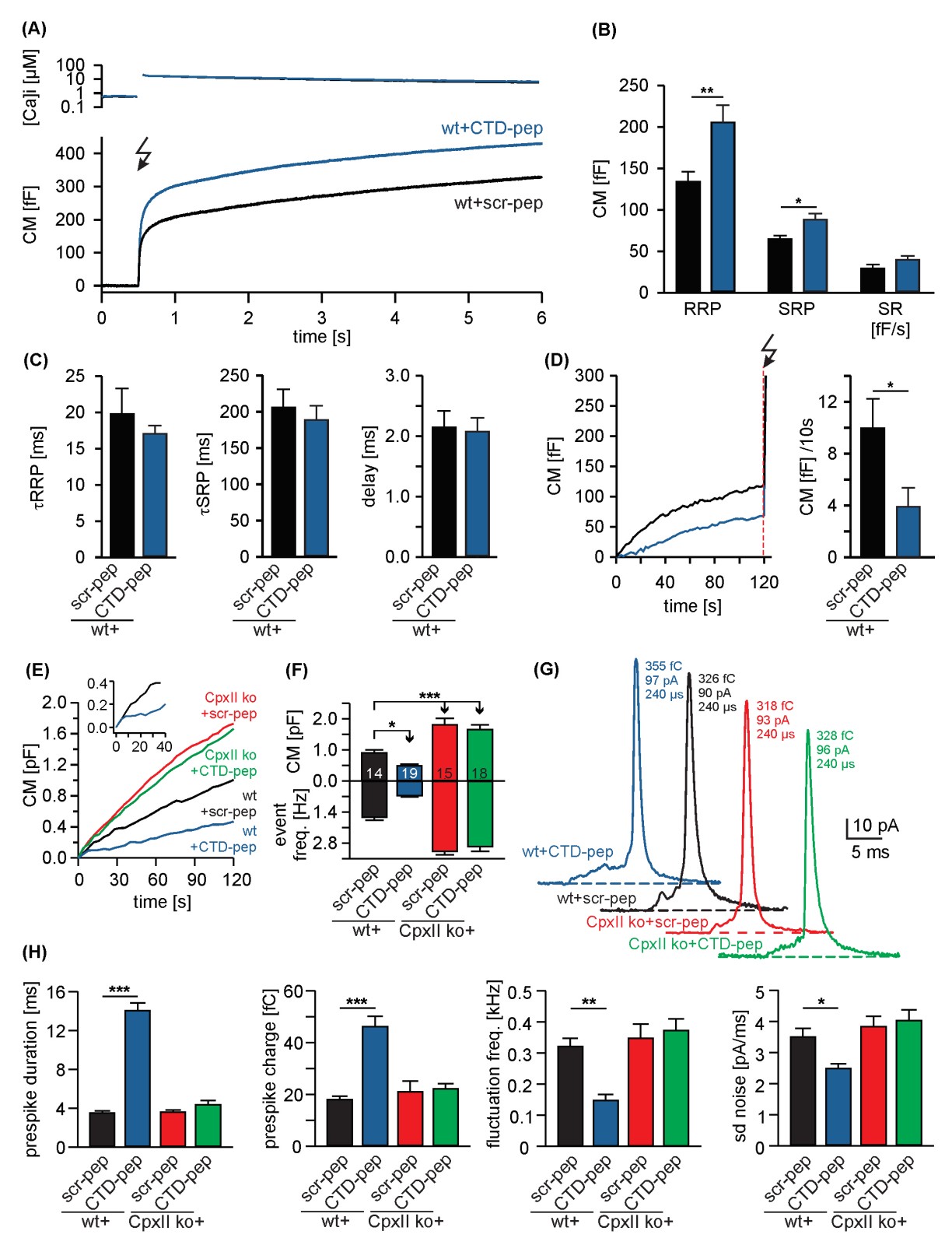

**Figure 2.** Acute infusion of CpxII-CTD inhibits premature exocytosis and boosts synchronous secretion. (**A**) Mean [Ca²⁺]i levels (top) and corresponding capacitance responses (bottom) of wt cells infused with either CpxII C-terminal peptide (wt + CTD pep, n = 29) or scrambled peptide (wt + scr pep, n = 29). Flash is at t = 0.5 s (arrow). (**B**) RRP and SRP size are significantly enhanced with CTD-pep infusion. (**C**) Neither the CTD-pep nor its scramble variant alters the kinetics of stimulus-secretion coupling (τRRP, τSRP and delay). (**D**) Mean tonic CM increase of the cells shown in A (left panel) before

*Figure 2 continued on next page*

*Figure 2 continued*
the UV-flash (arrow). Mean rate of ΔCM over 120 s (right panel, Mann Whitney U-test versus control). (**E–H**) CpxII-CTD is essential but not sufficient for clamping tonic release and early fusion pore. (**E**) The CTD-pep reduces tonic secretion (upon infusion with 19 μM [Ca$^{2+}$]i) in wt cells, but fails to do so in CpxII ko cells (t = 0 is the time point about 10–15 s after establishing the whole-cell configuration). The inset displays the initial CM response of wt cells infused with scr-pep (black) or CTD-pep (blue) at the expanded time scale. (**F**) Total ΔCM after 120 s (upper panel) and amperometric event frequency (lower panel, numbers indicate averaged cells). (**G**) Exemplary single vesicular release events (with similar charge and 50–90% rise time) illustrate the CTD-mediated increase in prespike duration only in wt cells (colour code as in E). (**H**) CTD-pep infusion significantly reduces fusion pore dynamics in wt cells but not in CpxII ko cells. Values are given as mean of median determined from the parameter's frequency distribution for each cell. Data are averaged from cells/events measured for wt + scr pep (14/1531), wt + CTD pep (19/889), CpxII ko+ scr pep (15/2692) and CpxII ko+ CTD pep (18/3052) (>20 events/cell). Error bars indicate mean ± SEM. ANOVA followed by Tukey-Kramer post-hoc test. *p<0.05, **p<0.01, ***p<0.001.
DOI: https://doi.org/10.7554/eLife.38981.005
The following figure supplement is available for figure 2:

**Figure supplement 1.** Infusion of CpxII C-terminal peptide slows the expansion of the initial fusion pore but leaves bulk release phase unchanged.
DOI: https://doi.org/10.7554/eLife.38981.006

In contrast to wt cells, infusion of the CTD-peptide in CpxII ko cells was unable to inhibit chromaffin granule exocytosis and failed to slow down the expansion of the initial fusion pore (*Figure 2E–H*), showing that the observed phenotype in wt cells is not simply a consequence of off-target effects with peptide infusion. These results rather suggest that other domains of CpxII either directly cooperate with the CpxII CTD or are required to bring the exocytotic machinery into a state wherein CpxII CTD can exert its inhibitory action. In any case, a physical continuity through the length of CpxII does not seem to be obligatory for the CTD function.

## N-terminal domains of CpxII cooperate with its far CTD in controlling exocytosis

To test whether separated CpxII domains can act in tandem to establish a fully functional ensemble, we virally expressed the N-terminal region of the protein (Cpx$^{1-100}$) in CpxII ko cells and infused the CTD-peptide via patch pipette (*Figure 3A*). Strikingly, infusion of CTD-peptide into Cpx$^{1-100}$ expressing CpxII ko cells promoted Ca$^{2+}$-evoked synchronous exocytosis almost like full-length CpxII expression. The scr-peptide, instead, did not restore synchronous release in Cpx$^{1-100}$ expressing cells (*Figure 3B,C*). The kinetic properties of the EB components remained unaffected for all tested groups (*Figure 3D*). Moreover, the Cpx$^{1-100}$ mutant together with the CTD-peptide inhibited tonic secretion as effectively as CpxII expression (*Figure 3E–H*), thus allowing for full restoration of the synchronous release component (*Figure 3B*). Analyses of single amperometric events revealed that the C-terminal peptide complements Cpx$^{1-100}$ mutant to decelerate the rate of initial fusion pore expansion and suppressed the fusion pore jitter like full-length CpxII (*Figure 3I*) without changing the properties of the main amperometric spike (*Figure 3—figure supplement 1*). Thus, separated N- and C-terminal regions of CpxII cooperate in forming a fully functional entity. These composite protein domains may arrest chromaffin granules at a 'primed' pre-fusion state and slow down fusion pore expansion impeding post-fusional cargo release.

We also tested a chimeric protein consisting of the Cpx$^{1-100}$ domain and the cysteine rich region of CSP-α protein (*Figure 3—figure supplement 2A*), which is expected to tether the fusion protein at the vesicle membrane (*Gong et al., 2016*). The Cpx$^{1-100}$CSPα mutant concentrated on LDCVs like the wt protein (for detailed analysis see *Figure 7—figure supplement 1*), but failed to thwart tonic vesicle fusion and did not support subsequent synchronous Ca$^{2+}$-triggered release in CpxII ko cells (*Figure 3—figure supplement 2B–F*). Moreover, expression of Cpx$^{1-100}$CSPα mutant in wt cells strongly suppressed EB component of evoked exocytosis (*Figure 3—figure supplement 2G–K*) compared to controls. Therefore, Cpx$^{1-100}$CSPα is functionally intact, competes with the endogenous protein, but is unable to clamp premature fusion and to build-up the release-ready vesicle pool. These results confirm that vesicular tethering of other Cpx domains by the CpxII CTD does not suffice to explain its inhibitory phenotype. Therefore, other interactions of the CTD are required to mediate the suppression of exocytosis by the protein.

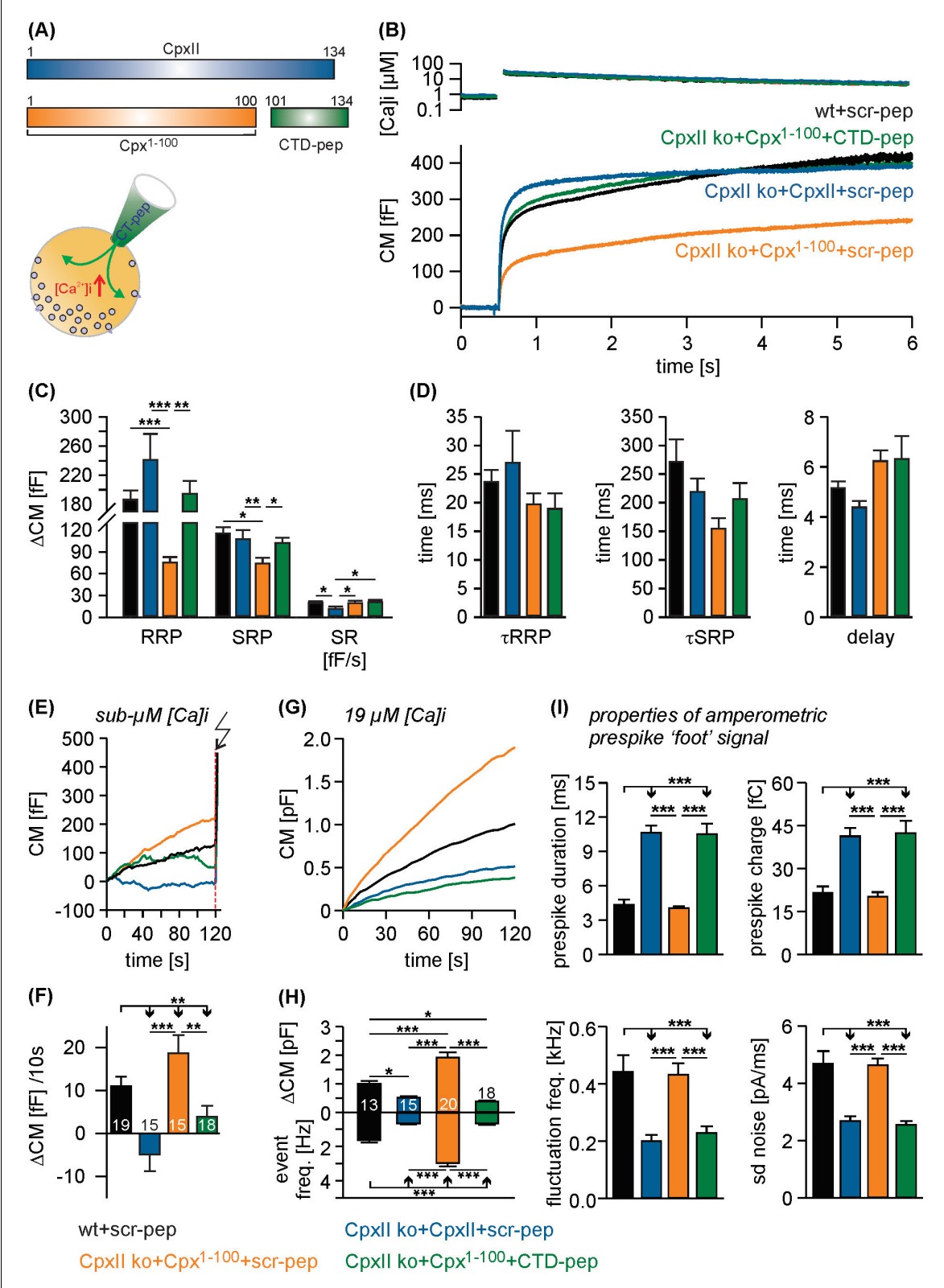

**Figure 3.** The CTD peptide and the Cpx[1-100] act in tandem to clamp tonic exocytosis and fusion pore expansion. (**A**) The CTD-pep (green) or scr-pep is infused into CpxII ko cells expressing either full-length CpxII (blue) or the truncated Cpx[1-100] (orange). (**B**) Average $[Ca^{2+}]_i$ levels (top) and corresponding capacitance responses (bottom) for infusion of either scr-pep or CTD-pep in the indicated groups. Data were collected from the following number of cells: wt + scr pep (black, n = 19), CpxII ko + CpxII + scr-pep (blue, n = 15), CpxII ko+Cpx[1-100]+scr pep (orange, n = 15) and CpxII
*Figure 3 continued on next page*

Figure 3 continued

ko+Cpx$^{1-100}$+CT pep (green, n = 18). Flash is at t = 0.5 s. (**C**) The CTD peptide (green), but not its scrambled variant (orange) recues the RRP and SRP in Cpx$^{1-100}$ expressing cells, matching nearly the phenotype of full length CpxII expression (blue). The rate of sustained release (SR, fF/s) is slightly reduced with CpxII expression. (**D**) Neither the time constants of the EB components (τRRP, τSRP) nor the exocytotic delay are altered for the tested groups. (**E, F**) Contrary to the scrambled peptide the CTD suppresses premature secretion at submicromolar [Ca]i before the flash response (arrow) in Cpx$^{1-100}$ expressing cells almost like the CpxII protein (same cells as shown in B). (**G**) Like CpxII expression in CpxII ko cells, the CTD hinders tonic secretion (in response to 19 µM [Ca]i) in Cpx1-100 expressing cells, whereas the scr-pep failed to do so. (**H**) Total CM after 120 s (upper panel) and amperometric event frequency (lower panel) for the groups in G. Numbers of cells are depicted in the bars. (**I**) The CTD-pep together with Cpx$^{1-100}$ increases the fusion pore expansion time (upper panels) and reduces its dynamics (lower panels) like CpxII. Values are determined from the individual parameter's frequency distribution for each cell. Data are averaged from the cells/events measured for wt + scr pep (13/1558), CpxII ko + CpxII + scr-pep (15/731), CpxII ko+Cpx$^{1-100}$ + scr pep (20/3292) and CpxII ko+Cpx$^{1-100}$+CTD pep (18/857) (>20 events/cell). Error bars indicate mean ± SEM. ANOVA followed by Tukey-Kramer post-hoc test. *p<0.05, **p<0.01, ***p<0.001.
DOI: https://doi.org/10.7554/eLife.38981.007

The following figure supplements are available for figure 3:

**Figure supplement 1.** Neither CpxII expression nor a combination of Cpx$^{1-100}$ expression and acute peptide infusion affects the main phase of transmitter discharge from single vesicles.
DOI: https://doi.org/10.7554/eLife.38981.008
**Figure supplement 2.** CSPα—anchored Cpx$^{1-100}$ fails to support secretion like CpxII.
DOI: https://doi.org/10.7554/eLife.38981.009

## The C-terminus of CpxII slows down ternary SNARE complex formation

The phenotypical alterations in nascent fusion pore dynamics with CpxII expression (*Figure 1—figure supplement 2*) are remarkably similar to those observed with sybII linker mutants designed to diminish the force transfer on to the merging membranes (*Kesavan et al., 2007*). Furthermore, CpxII clamps tonic secretion even in the absence of the major Ca$^{2+}$-sensors Synaptotagmin I (SytI) and Synaptotagmin VII in chromaffin cells (*Dhara et al., 2014*). These functional observations point to the possibility that CpxII may directly hinder progressive SNARE assembly, arresting the complex in a partially zippered state. To probe for putative protein-protein interactions, we incubated the CpxII-CTD peptide or its scrambled variant (immobilized on sulfo-link sepharose beads) detergent extract of mouse brain homogenate. Western blot analyses revealed that CTD-peptide co-isolates significant amounts of SytI, Syx-1A, SNAP25 and traces of SybII, but showed no binding to proteins like synapsin1 (*Figure 4A*). Furthermore, with the exception of low amounts of SytI binding, none of the other protein interactions was observed with the scrambled peptide. These results agree with a previous report implicating the glutamate cluster of the Cpx C-terminus (amino acid 108–114) in binding to SytI (*Tokumaru et al., 2008*). They further suggest that, in addition to the well-known interaction between the CpxII central α helix and the SNARE complex (*Chen et al., 2002*), the CpxII CTD provides an alternative SNARE binding motif. To further substantiate this hypothesis, we studied whether the C-terminus of CpxII affects the kinetics of SDS-resistant SNARE complex formation. By using purified recombinant SNARE proteins, we assayed assembly kinetics with SDS page and Coomassie staining after defined time intervals (*Figure 4B*). The kinetics of complex formation were reasonably well approximated by second order reaction kinetics (*Nicholson et al., 1998*; *Pobbati et al., 2006*). The experiments revealed that the CTD-peptide significantly decelerated the rate of SNARE complex assembly (51.4 ± 3.8% of control, p<0.05, ANOVA on ranks with Dunn's post test), when compared with scr-peptide (106.6 ± 16.7% of control) or no peptide addition (control) (*Figure 4C*), whereas the overall amount of assembled SNARE complexes determined after 240 min was unchanged (% of control: CTD-peptide 92.3 ± 9.1, scr-peptide 88.8 ± 9.6, p>0.9). Thus, the CpxII C-terminus transiently hinders the assembly of SNARE proteins.

Previous studies have suggested, that SNARE complex formation is most likely arrested halfway leaving membrane-proximal layers of the SNARE bundle unzippered and free for other potentially competing protein-protein interactions (*Gao et al., 2012*; *Giraudo et al., 2006*; *Hernandez et al., 2012*; *Li et al., 2014*; *Li et al., 2016*; *Pobbati et al., 2006*; *Zhou et al., 2017*). While CpxII positions with its central helix on the assembling SNARE complex in an antiparallel orientation, downstream protein regions of CpxII may provide sufficient structural flexibility for the far C-terminus to fold back on membrane-proximal layers of the partially assembled SNARE complex. Indeed, single molecule FRET studies have provided evidence that Cys105 within the Cpx CTD interacts with Syx 1a

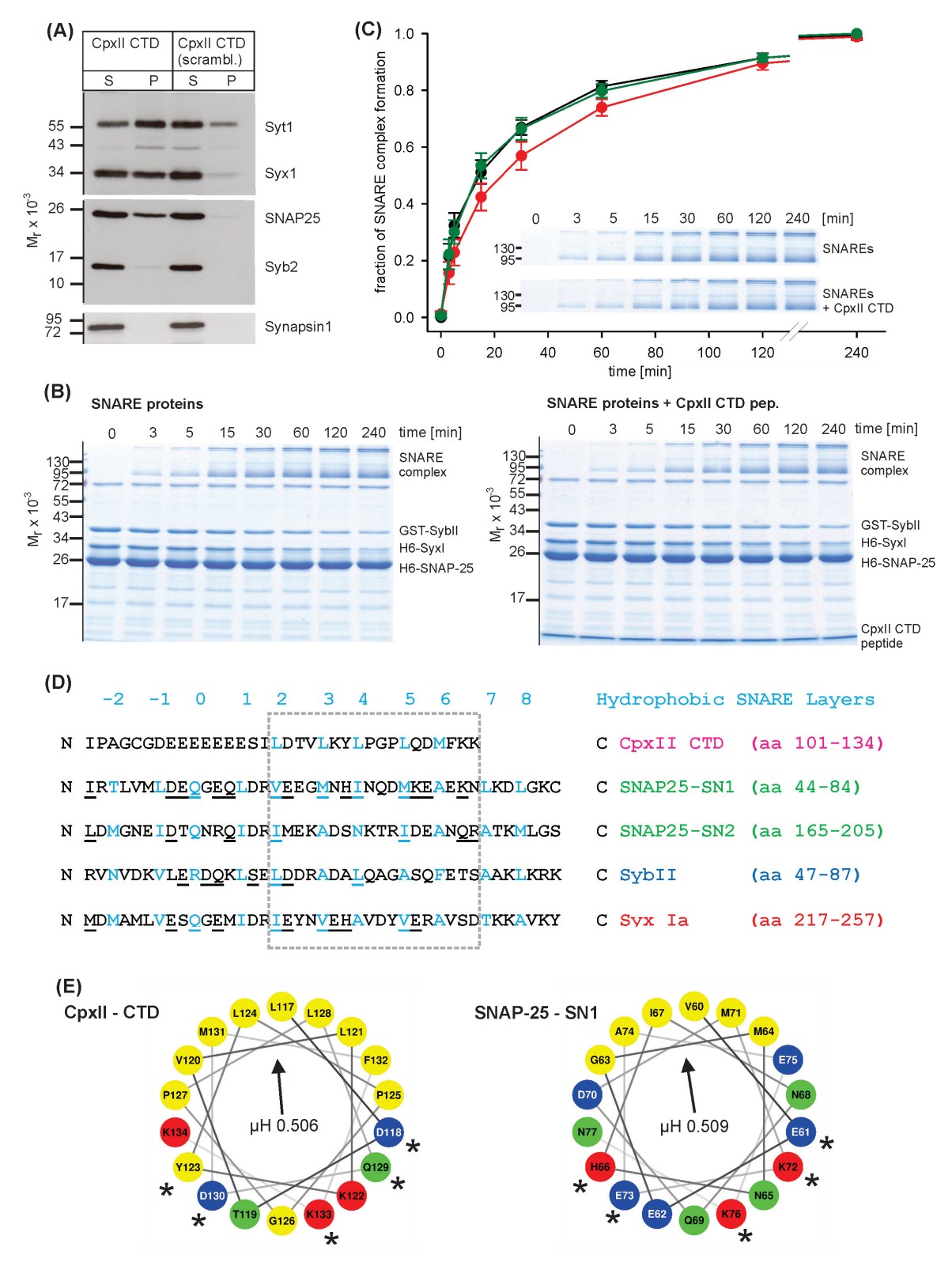

**Figure 4.** The CpxII CTD slows down the rate of SNARE complex formation. (**A**) The CpxII CTD peptide co-precipitates SytI and preferentially the t-SNAREs, Syx1a and SNAP25, from detergent extract of mouse brain. Equal volumes of supernatant (S) and pellets (P) were analyzed by SDS-PAGE (12% gel) and Western blotting with antibodies against the indicated antigens. No binding of other proteins like synapsin could be detected. The scrambled variant binds unspecifically some SytI and Syx1. (**B**) Time-dependent SNARE complex formation between GST-syb2 (3 µM) and preincubated

*Figure 4 continued on next page*

*Figure 4 continued*

t-SNAREs (syntaxin 1, amino acids 1–262, 3 µM and SNAP25, amino acids 1–206, 15 µM) in the absence (left) and the presence (right) of the CpxII-CTD (50 µM). Complex formation was determined at 25°C for the indicated times and analyzed by SDS-PAGE. Exemplary Coomassie-stained SDS gels are shown. (C) The CpxII CTD peptide (red) slows down the time course of SNARE complex formation when compared with no peptide (SNAREs alone, black) or the scrambled peptide (green). Inset highlights delayed complex assembly in the presence of the CpxII CTD peptide (same gels as shown in B). Data were normalized to the complex formation after 240 min and averaged from the following number of trials (SNAREs, n = 8; SNAREs + CpxII CTD, n = 8; SNAREs + scr pep, n = 7). (D) Sequence alignment between CpxII CTD and the SNARE motifs of SNAP25, SybII and Syx1 (hydrophobic layer region −2 to +8, blue). The CpxII-CTD shows a high degree of similarity (underlined residues) to SNAP25-SN1 (50%, calculated for the boxed region, hydrophobic layers + 2 to7, using the BLOSUM62 matrix). (E) Helical wheel projections of CpxII CTD (residues 117–134, boxed region in D) and SNAP25-SN1 (residues 60–77) show the amphipathic nature of the protein regions with similar hydrophobic moments (µH). Similar amino acids in the hydrophilic faces of the helices are marked with asterisks (*).

DOI: https://doi.org/10.7554/eLife.38981.010

near the ionic (0) layer of the SNARE complex (*Bowen et al., 2005*). Given the promiscuity of hydrophobic SNARE interactions (*Tsui and Banfield, 2000*; *Yang et al., 1999*), the CpxII CTD with its hydrophobic residues (presented in heptad repeats) may associate with the membrane-proximal layers of the SNARE proteins and interfere with their zippering. Sequence comparison between the CpxII C-terminus and membrane-proximal layers of the SNAREs revealed stronger similarities with the SNARE motif of SNAP25a SN1 (hydrophobic layers + 2 to7, *Figure 4D*; 50% similarity, see Materials and methods) than with any other attempted alignment of this protein region with either SNAP25a SN2, Syx 1A or SybII (27.8%, 38.9% and 16.7% similarity, respectively). Helical wheel presentations of CpxII-CTD (residues 117 to 134) and SNAP25-SN1 (residues 60 to 77) illustrate the amphiphilicity of both helices with similarly angled, nearly identical hydrophobic moments (µH: CTD 0.506, SN1 0.509, *Figure 4E*) – a parameter that is disparate for the corresponding regions within SNAP25-SN2 (µH: 0.372), SybII (µH: 0.231) and Syx1a (µH: 0.533, boxed residues in *Figure 4D*). Furthermore, sequence comparisons of the Cpx C-terminal regions from different species (i.e. *Drosophila melanogaster, Loligo pealeii, Hirudo medicinalis*) with their corresponding SNARE proteins confirm the view that the Cpx C-terminus is on average most similar to the SNAP25-SN1 domain (in %: SNAP25-SN1, 41.6 ± 3.5; SNAP25-SN2, 27.7 ± 2.2; Syx, 34.7 ± 8.2; SybII, 19.4 ± 1.5). This suggests functional similarities of vertebrate and invertebrate complexins and a high degree of intraphylum conservation with respect to the primary sequence and the amphipathic sequence pattern. Collectively, our functional data, biochemical results, and sequence comparisons point to the possibility that the CpxII C-terminus with its SN1 mimetic region may compete with the SN1 motif (membrane-proximal layers) of SNAP25 for binding to its cognate SNARE partners, thereby hindering SNARE assembly and effectively arresting exocytosis.

## A SNAP25-SN1 C-terminus restores function to the CpxII protein

To test whether the CpxII C-terminus and the SNAP25-SN1 domain are functionally interchangeable, we generated a chimera protein where the last 34 amino acids of CpxII (corresponding to the CTD-peptide) are replaced with the equivalent region of SNAP25-SN1 (residues 44 to 77, see *Figure 4F*). Strikingly, the CpxII:SN1 chimera (Cpx$^{1-100}$-Long Chimera, also referred to as Cpx$^{1-100}$LC, *Figure 5A*) largely restored the magnitude and the kinetics of synchronous exocytosis (*Figure 5B–E*) and fully suppressed tonic vesicle fusion at similar sub-micromolar [Ca$^{2+}$]i concentrations prior to the flash induced secretion response. (*Figure 5F–H*). To study whether any unrelated alpha helical domain can serve as a substitute for the CpxII CTD, we replaced the C-terminus of CpxII with an artificial alpha helical domain (*Radoff et al., 2014*) formed by multiple repeats of Glu-Ala-Ala-Lysine sequences (A-(EAAK)$_8$-A (Cpx$^{1-100}$helix, *Figure 5A*). The Cpx$^{1-100}$helix mutant largely failed to increase the flash-evoked secretion response beyond the level of the corresponding Cpx$^{1-100}$ truncation mutant (compare *Figure 5B,D* and *Figure 1A*). Yet, this mutant rescued the kinetics of stimulus secretion coupling (τRRP, τSRP) and the secretory delay, as they are mediated by an intact N-terminal region of CpxII (*Dhara et al., 2014*). Furthermore, the Cpx$^{1-100}$helix was also unable to clamp 'pre-flash' release, remaining close to the level of the CpxII ko (*Figure 5F–H*).

Replacing the amphipathic region within the CpxII C-terminus (last 19 amino acids) with residues 59 to 77 of SNAP25-SN1 (Cpx$^{1-115}$-Short Chimera, also referred to as Cpx$^{1-115}$SC, *Figure 5—figure supplement 1A*) even fully restored the functionality of CpxII protein. Indeed, the Cpx$^{1-115}$SC with

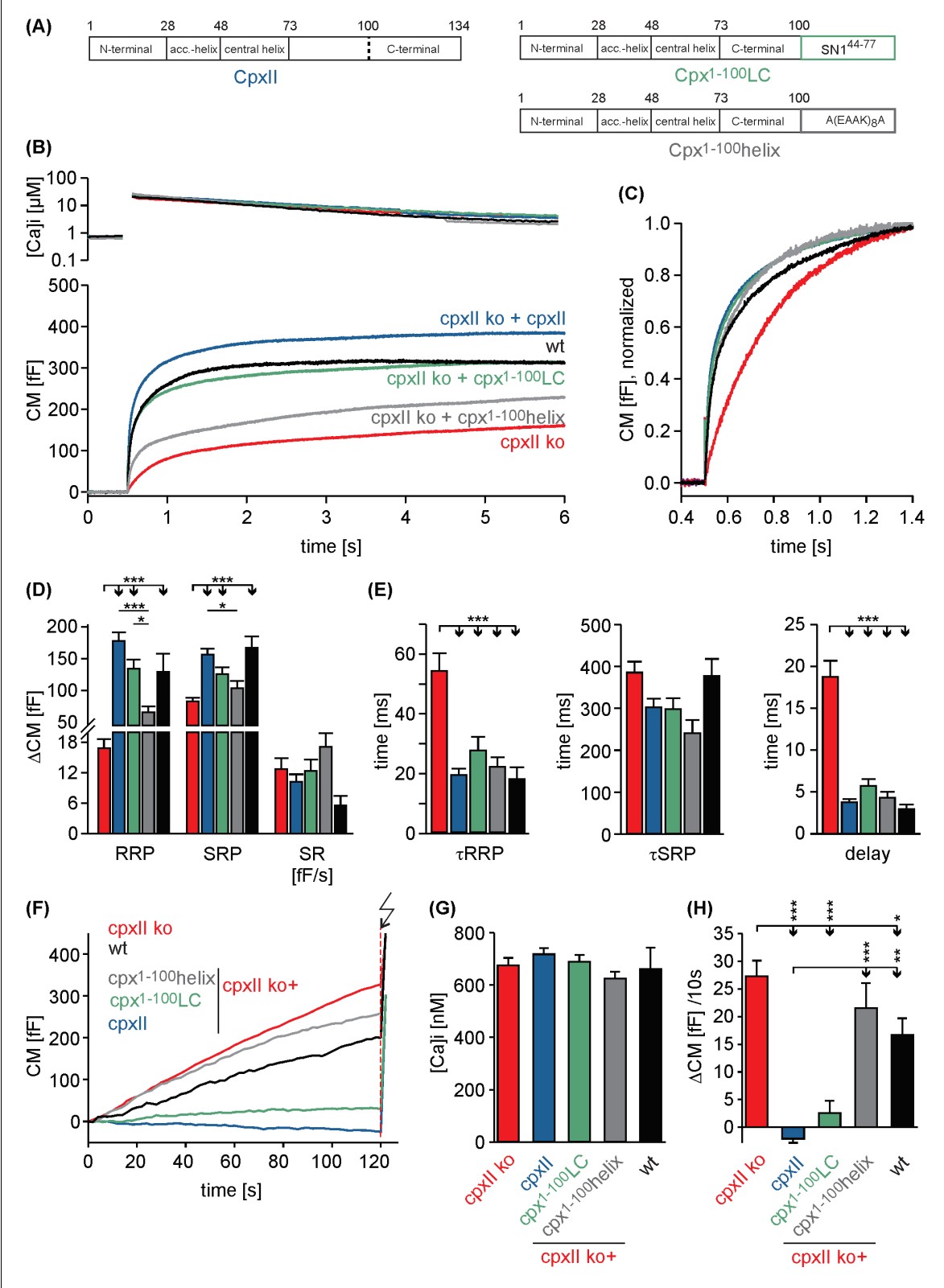

**Figure 5.** Replacing the CpxII C-terminus with SNAP25-SN1 (residues 44 to 77) largely restores synchronous exocytosis. (**A**) Cartoon illustrates CpxII domains and the structure of the tested mutants. (**B**) Expression of the Cpx$^{1-100}$LC largely restores synchronous secretion reaching the response level of wt cells. The Cpx$^{1-100}$helix supports exocytosis like Cpx$^{1-100}$ (compare with **Figure 1B**). Mean [Ca$^{2+}$]i levels (top) and corresponding CM signals (bottom) of wt cells (n = 11), CpxII ko cells (n = 58), and ko cells expressing CpxII (n = 68) or its mutants (Cpx$^{1-100}$LC, n = 41; Cpx$^{1-100}$helix, n = 20). Flash, t = 0.5

*Figure 5 continued on next page*

Makke *et al*. eLife 2018;7:e38981. DOI: https://doi.org/10.7554/eLife.38981                                        11 of 25

Figure 5 continued

s. (C) Normalized CM (as shown in B) scaled to the wt response (1 s after flash). (D) Cpx$^{1-100}$LC (green) restores RRP and SRP almost like CpxII (blue). (E) The time constants of the EB components (τRRP and τSRP), and the exocytotic delay are similarly changed for CpxII and its chimera (color coding as in A). (F–H) The Cpx$^{1-100}$LC clamps tonic secretion (F) of CpxII ko cells at similar [Ca]i (G) almost like CpxII (F, H). ANOVA followed by Tukey-Kramer post-hoc test. *p<0.05; **p<0.01; ***p<0.001. Error bars indicate mean ± SEM.
DOI: https://doi.org/10.7554/eLife.38981.011
The following figure supplement is available for figure 5:

**Figure supplement 1.** The Cpx$^{1-115}$SC arrests tonic release and fully rescues synchronous secretion like CpxII.
DOI: https://doi.org/10.7554/eLife.38981.012

its SNAP25-matching C-terminus supports undiminished synchronized secretion (*Figure 5—figure supplement 1B–F*) and clamps premature tonic release like CpxII (*Figure 5—figure supplement 1G,H*). In contrast, the corresponding artificial helix chimera (last 19 residues replaced with (EAAK)$_4$-EAA, Cpx$^{1-115}$helix) again behaved like the truncation mutant (Cpx$^{1-115}$). Thus, long and short chimera constructs of CpxII:SN1 arrest premature vesicle fusion and recover the 'primed' vesicle pool.

Simultaneous recordings of membrane capacitance and amperometric spike frequency confirmed that expression of Cpx$^{1-100}$LC or Cpx$^{1-115}$SC in CpxII ko cells strongly hindered tonic secretion as did the wt protein (*Figure 6A,B*). Quantification of amperometric prespike properties revealed that Cpx$^{1-100}$LC and Cpx$^{1-115}$SC reduced the fusion pore expansion rate and inhibited current fluctuations during the prespike signal as efficiently as CpxII (*Figure 6C–E*). In contrast, the Cpx$^{1-100}$helix mutant not only failed to clamp tonic vesicle fusion, but also did not affect the kinetics of initial fusion pore (*Figure 6*). Neither CpxII nor the mutant variants changed the main spike phase of the amperometric events with respect to their charge, amplitude, rise time and half width (*Figure 6—figure supplement 1*), agreeing well with our previous observations (*Dhara et al., 2014*). Furthermore, immunofluorescence analyses showed that the various mutants of CpxII were expressed with similar efficiency as the wt protein (*Figure 6—figure supplement 2*). Thus, any difference in protein expression cannot be held responsible for the contrasting clamping ability observed with the tested mutants. Taken together, by using a variety of functional analyses, we find that chimera proteins with a SNAP25 C-terminus fully support CpxII action in Ca$^{2+}$-triggered exocytosis - not only in exocytosis synchronization and timing, but also during transmitter discharge from single vesicles.

## The CpxII$^{1-100}$LC chimera 'superclamps' the exocytotic machinery

Given the SNAP25 C-termini of Cpx$^{1-100}$LC and Cpx$^{1-115}$SC, one might speculate that such mutants should even enhance the clamp action of CpxII. Yet, high levels of virally driven protein expression (*Figure 6—figure supplement 2*) may have occluded a 'gain-of-function' phenotype for the mutant protein compared to wt CpxII. To address this issue, we have reduced viral protein expression to endogenous CpxII levels by lowering the concentration of infectious particles (25%) and shortening the expression time (3.5 hr instead of 5.5 hr, *Figure 7A*). Under these conditions, immunofluorescence analyses revealed nearly similar expression levels of CpxII and mutant proteins in CpxII ko cells when compared to those of the endogenous protein in wt cells. In good agreement, the electrophysiological analyses showed that expression of CpxII in CpxII ko cells reduced tonic exocytosis only to levels observed in wt cells (*Figure 7B*, compare with *Figure 6A*). Furthermore, the CpxII$^{1-100}$LC mutant with its C-terminal SNAP25-SN1 domain hindered both, tonic secretion and fusion pore expansion more strongly than CpxII (*Figure 7B–E*, compare with *Figure 6D,E*). Thus, SNAP25 structures within the CpxII C-terminus significantly enhance the clamp action of the CpxII protein, fostering a 'superclamp' phenotype. By taking advantage of these adapted protein expression levels, we also investigated the question whether a potential vesicle association of the CpxII$^{1-100}$LC can be held responsible for its unperturbed clamp function. For this, we probed the subcellular distribution of CpxII in chromaffin cells using SybII as a LDCV marker protein (*Dhara et al., 2016*). The subcellular distribution of the chromaffin granules was determined with confocal microscopy at the footprint of the cell. The CpxII staining in wt cells showed a clear punctate appearance that often coincided with sybII positive granules (*Figure 7—figure supplement 1A*). This indicates a vesicular concentration of the protein, an observation that was further corroborated by line-scan analyses of sybII fluorescent puncta and a high Pearson's coefficient (0.59 ± 0.02, n = 17), determined for the entire cell area, *Figure 7—figure supplement 1A,F,G*). While expression of CpxII in CpxII ko cells

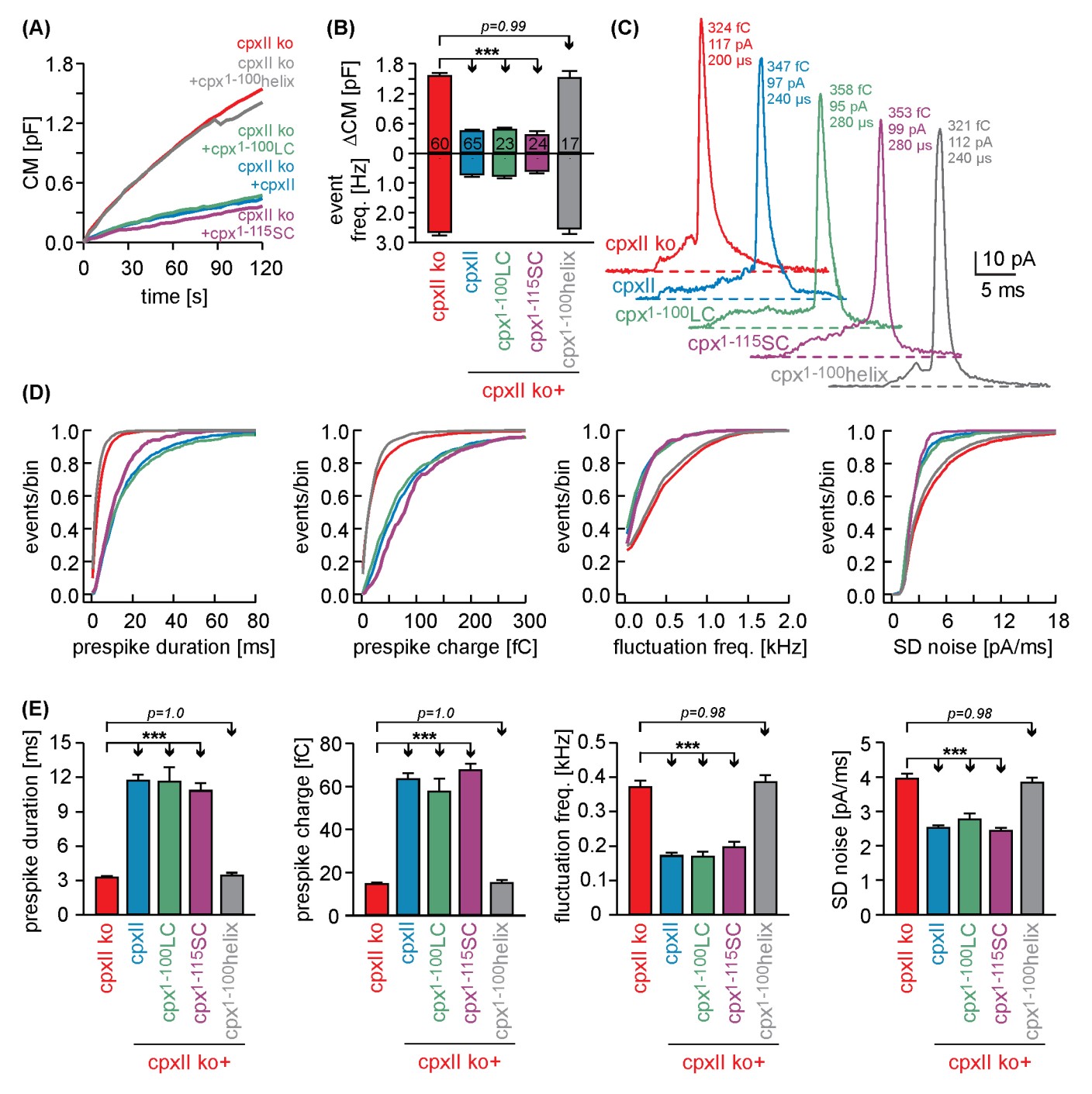

**Figure 6.** Cpx$^{1-100}$LC and Cpx$^{1-115}$SC hinder tonic secretion and regulate fusion pore dynamics like CpxII. (**A**) Mean CM responses upon intracellular perfusion with 19 µM [Ca$^{2+}$]i for the indicated groups. (**B**) Corresponding ΔCM (determined after 120 s, top panel) and amperometric event frequency (bottom panel, number of cells denoted in the bars). (**C**) Exemplary single vesicular release events illustrate the slower fusion pore expansion time for CpxII and the CpxII:SNAP-25 chimeras. (**D, E**) Cumulative frequency distributions (D, color coding as in A) and cell-weighted averages (**E**) of the indicated prespike parameters show that Cpx$^{1-100}$LC and Cpx$^{1-115}$SC prolongs the fusion pore expansion time and reduces fusion pore jitter as seen with CpxII. Values are determined from the individual parameter's frequency distribution for each cell. Data are collected from events/cells measured for CpxII ko (10735/60), ko + CpxII (3842/65), ko+Cpx$^{1-100}$LC (1143/23), ko+Cpx$^{1-115}$SC (1192/24) and ko+Cpx$^{1-100}$helix (2873/17) (>20 events/cell). Error bars indicate mean ± SEM. ANOVA followed by Tukey-Kramer post-hoc test. ***$p<0.001$.

DOI: https://doi.org/10.7554/eLife.38981.013

The following figure supplements are available for figure 6:

*Figure 6 continued on next page*

*Figure 6 continued*

**Figure supplement 1.** Chimera mutants of CpxII do not affect the main phase of transmitter discharge from single vesicles.
DOI: https://doi.org/10.7554/eLife.38981.014
**Figure supplement 2.** Expression analysis of full-length CpxII and its mutant proteins.
DOI: https://doi.org/10.7554/eLife.38981.015

exhibited a similar subcellular distribution, truncation of the CpxII CTD (Cpx$^{1-100}$) abolished the vesicular accumulation of the protein, agreeing well with previous studies in vitro and in vivo (*Gong et al., 2016*; *Malsam et al., 2012*; *Wragg et al., 2013*; *Zdanowicz et al., 2017*). The CpxII: SN1 chimeras exhibited a similar diffused distribution throughout the cell cytoplasm (*Figure 7—figure supplement 1D–G*), showing that vesicular binding of the C-terminus is not a prerequisite for the clamp action of the protein. Furthermore, the Cpx$^{1-100}$CSPαmutant accumulated on LDCVs like the wt protein (*Figure 7—figure supplement 1C,G*), but entirely failed to hinder tonic vesicle fusion (*Figure 3—figure supplement 2*), indicating that vesicular localization of Cpx is not sufficient for its inhibitory function. Taken together, these results support the view that protein-protein interactions of CpxII's C-terminus, most likely by hindering SNARE complex assembly, rather than its protein-lipid interactions are instrumental in arresting vesicle exocytosis.

## Discussion

Complexin is the predominant inhibitor at many synapses and neuroendocrine cells to suppress untimely fusion and to sustain a pool of release-ready vesicles. Yet, the precise mode of CpxII action has remained controversial. Here, we show that the CTD of CpxII maintains tight control over premature vesicles exocytosis to support a pool of release-ready vesicles. Our experiments demonstrate that crucial inhibitory actions of CpxII's CTD are not due to membrane binding or vesicle association, but are mediated by its SNAP25-SN1 mimetic properties. In close correlation, we show that CpxII chimera with a SNAP25-SN1 C-terminal domain fully restores the protein's clamp action and its ability to support synchronous neurotransmitter release. Collectively, our results deliver new insight into fundamental mechanisms that constitute the molecular clamp of Ca$^{2+}$-triggered exocytosis. They support a model wherein the CpxII C-terminus hinders SNAP25 from binding to its cognate SNARE partners, arrests the complex in a partially-zippered state and thus allows for highly synchronized Ca$^{2+}$-triggered release.

### Mechanisms of CpxII clamp action in exocytosis

Previous results by Rothman and colleagues identified a region comprising amino acids 26–83 of murine Cpx as the 'minimal clamping domain' of the protein (*Giraudo et al., 2006*; *Giraudo et al., 2009*). Based on a comprehensive mechanistic model, it has been suggested that the accessory helix of Cpx blocks the binding of membrane proximal parts of SybII, thereby, preventing full zippering of the SNARE complex. While attempts to test this model by generation of Cpx-mutants with enhanced or decreased sequence similarities to SybII revealed the expected results in vitro (*Giraudo et al., 2009*), in vivo studies could demonstrate only mild effects or produced even inconsistent results regarding the efficacy of the mutant proteins to either 'superclamp' or 'unclamp' spontaneous release (*Cho et al., 2014*; *Trimbuch et al., 2014*; *Yang et al., 2010*). Moreover alternative models have been proposed wherein the accessory alpha helix inhibits release by electrostatic repulsion with the vesicle membrane (*Trimbuch et al., 2014*) or by helix propagation into the central α-helical domain of Cpx (*Radoff et al., 2014*). These opposing views and also the disparate results from in vitro and in vivo studies point to the possibility that alternative modes for Cpx's clamp action do exist and assist to stall unfettered vesicle fusion. Experimental results from our group and others have previously shown that the CTD of CpxII exerts a fusion clamping function, although the underlying mechanisms remained largely enigmatic (*Buhl et al., 2013*; *Dhara et al., 2014*; *Kaeser-Woo et al., 2012*; *Martin et al., 2011*). Furthermore, experiments at the NMJ of C. elegans and murine cortical neurons suggested that the CTD of Cpx guides the protein to vesicular membranes in a curvature sensitive fashion and thus concentrates other inhibitory domains of CpxII at the site of exocytosis for fusion clamping (*Gong et al., 2016*; *Snead et al., 2017*; *Snead et al., 2014*; *Wragg et al., 2017*; *Wragg et al., 2013*). Several lines of evidence presented in this paper counter

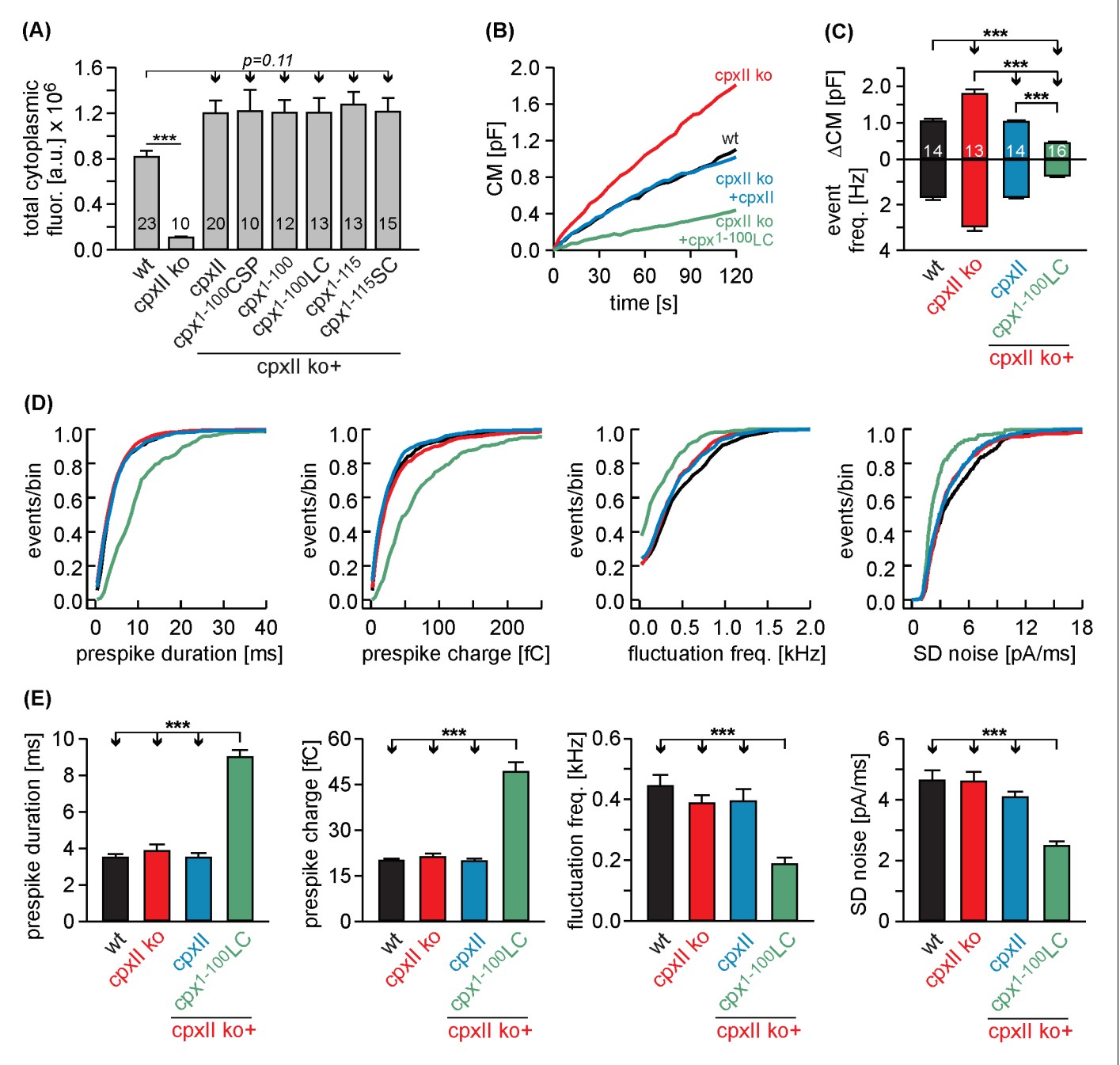

**Figure 7.** The Cpx$^{1-100}$LC 'superclamps' tonic secretion and fusion pore dynamics after adapting the expression to endogenous CpxII levels. (A) Mean total fluorescence intensity for wt and CpxII ko cells expressing either CpxII or the indicated mutants (measured 3 hr after transfection, number of cells is indicated in the bars). CpxII and its mutant variants show similar protein expression levels compared to endogenous CpxII. (B) Mean CM responses upon intracellular perfusion with 19 μM free Ca$^{2+}$ for the indicated groups. Note that Cpx$^{1-100}$LC hinders tonic secretion more efficiently than CpxII (compare with *Figure 6A*). (C) Total ΔCM after 120 s (top panel) and amperometric event frequency (bottom panel) averaged from the indicated number of cells. (D, E) Properties of the prespike foot signal, displayed as cumulative frequency distribution (D) and as cell weighted averages (E) for the indicated parameters. The Cpx$^{1-100}$LC selectively slows the expansion rate of initial fusion pore and reduces the prespike kinetics compared to controls. Values are determined from the individual parameter's frequency distribution for each cell. Data are collected from events/cells measured for wt (1072/14), CpxII ko (1726/13), ko + CpxII (1125/14), ko+Cpx$^{1-100}$LC (600/16) (>20 events/cell). Error bars indicate mean ± SEM. ANOVA followed by Tukey-Kramer post-hoc test. ***p<0.001.

DOI: https://doi.org/10.7554/eLife.38981.016

The following figure supplement is available for figure 7:

*Figure 7 continued on next page*

*Figure 7 continued*

**Figure supplement 1.** The CTD concentrates CpxII on chromaffin granules.
DOI: https://doi.org/10.7554/eLife.38981.017

the hypothesis that the CTD simply targets CpxII to membrane, but rather suggest a direct blockade of SNARE zippering. First, we show that the Cpx[1-100] mutant (lacking the last 34 amino acids) impairs synchronized secretion in wt cells and disinhibits tonic secretion to the level of the CpxII ko phenotype. Thus, the truncated mutant protein, lacking the amphipathic helix for membrane targeting successfully competes with endogenous CpxII for binding to the exocytotic machinery, but has lost its ability to hinder premature vesicle fusion. Second, acute infusion of the isolated C-terminal peptide into wt cells significantly diminishes premature vesicle fusion and enhances the subsequent phase of synchronized release as does expression of full-length CpxII. In the same line, our complementation experiments illustrate that separate N-and C-terminal domains of CpxII can efficiently reconstitute the inhibitory phenotype of the protein in CpxII ko cells. These results contrast the idea that vesicular targeting of CpxII by its CTD is essential for fusion clamping in chromaffin cells and indicate that the far CTD acts as an independent inhibitory module within the fusion machinery. Third, a Cpx[1-100]CSPα chimera (with CSPα serving as vesicular membrane anchor) is efficiently sorted to chromaffin granules, but completely fails to reinstate the inhibition by the protein. Although this mutant acts in dominant-negative fashion in wt cells implying efficient binding to productive SNARE complexes, it has lost its ability to arrest vesicle fusion. Collectively, these results are difficult to reconcile with a vesicle targeting role of the CTD of CpxII and rather suggest that this protein region plays an active role in fusion inhibition. Clearly, our observations do not exclude the possibility that vesicular localization is still functionally relevant as it concentrates CpxII at the sites of vesicle fusion (*Figure 7—figure supplement 1A,G*), but they also show that other modes of Cpx action dominate its inhibitory mechanism.

## A new model for the clamp function of CpxII C-terminus

As an alternative model, we propose that the CTD of CpxII with its amphipathic helix hinders SNARE complex assembly. The intramolecular distance between the central domain of CpxII and the far CTD appears to be long enough for the latter to fold back on membrane proximal parts of the SNARE complex and thus may hinder its further zippering (*Figure 8*). This notion is further supported by the observation that Cpx Cys105 positioned near the central ionic layer of the SNARE complex, leaving the downstream CTD region free to interact with the C-terminal layers of SNARE complex (*Bowen et al., 2005*). Our experiments support such a model by independent lines of evidence. The biochemical experiments show that the CpxII CTD (immobilized on agarose beads) coprecipitates Syt1, Syx1a, SNAP25, and some SybII from detergent extract of mouse brain homogenate. Given the high degree of structural similarity between the CpxII CTD and the C-terminal half of the SNAP25-SN1, it is possible that the amphipathic helix of CpxII CTD generates an alternative hydrophobic interface that competes with one of the SNARE motifs for complex formation. Since CpxII binds to binary as well as ternary complexes with high affinity (*Zdanowicz et al., 2017*), it remains to be shown at which particular step the protein interferes with SNARE assembly. In any case, we find that the CTD of CpxII, but not its scrambled variant, significantly lowers the rate of SDS-resistant SNARE complex formation. These in vitro results agree with previous reports, showing that full-length Cpx inhibits SNARE-mediated liposome fusion, whereas a truncated variant (amino acid 26–83) failed to do so (*Chicka and Chapman, 2009*; *Schaub et al., 2006*). Furthermore, and most strikingly, our functional analyses reveal that a corresponding chimera between CpxII and SNAP25-SN1 fully restores functionality regarding the magnitude of synchronized secretion, exocytosis timing and the expansion rate of the nascent fusion pore. In contrast, a length-matched substitution of the CTD of CpxII with an artificial alpha helix is unable to rescue secretion. While there are many ways to cripple a protein, only few are able to restore or even enhance its function. Our results indicate functional interchangeability between the CpxII CTD and the SNAP25-SN1 domain, rendering the possibility likely that the CTD of CpxII transiently interferes with SNARE complex assembly, generating a prefusion intermediate that can be activated by SytI in response to intracellular calcium rise. Importantly, using different experimental strategies with CTD-peptide infusion or CpxII:SN1

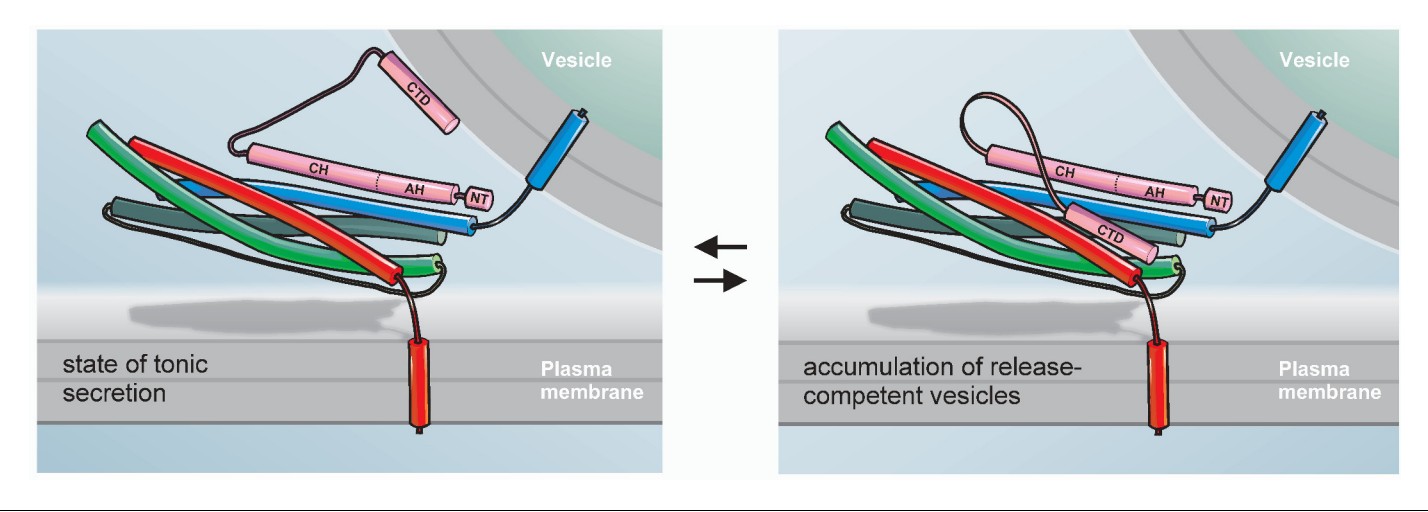

**Figure 8.** Hypothetical model for the exocytotic arrest by the CTD of CpxII. SybII (blue), Syx (red), and SNAP-25 (green) partially assemble into trans-SNARE complex forming a high affinity binding site for the central domain (CH) of Cpx (pink). In the membrane associated or free state of the CTD (left panel) CpxII does not inhibit tonic secretion, whereas in the SNARE-interacting state (right panel) the CpxII CTD folds back on the partially assembled SNARE complex, hinders its further zippering and thereby allows for the accumulation of release ready vesicles (adapted from *Mohrmann et al., 2015*). Halting the SNARE machinery before the triggering Ca$^{2+}$-stimulus most likely requires the combined action of multiple peptide domains from Cpx (e.g. accessory α-helix, AH, [*Giraudo et al., 2009*]) as well as other proteins like SytI (*Littleton et al., 1994*; *Martin et al., 1995*). The N-terminus of Cpx (NT) enhances fusion kinetics and fusogenicity (*Dhara et al., 2014*; *Xue et al., 2010*).

DOI: https://doi.org/10.7554/eLife.38981.018

chimera expression we find not only inhibition of tonic release as one might expect from SNAP25-mimetic structures (*Apland et al., 1999*; *Ferrer-Montiel et al., 1998*; *Gutiérrez et al., 1995*; *Gutierrez et al., 1997*), but also enhanced synchronized exocytosis indicating a true gain-of-function phenotype of the CTD of CpxII. In this context, it is important to emphasize that other chimera proteins of Cpx were often unsuccessful to reinstate functionality. For example, a chimeric construct between CpxI and the CpxIII CTD failed to suppress spontaneous synaptic vesicle release in murine cortical neurons (*Kaeser-Woo et al., 2012*), an observation that could be explained by the lack of periodicity of hydrophobic residues (heptad repeats) within the amphipathic helix of CpxIII. Furthermore, chimeric mutants between worm CpxI and the murine CpxI or CpxIII CTDs did not preserve a similar periodicity of hydrophobic residues and therefore may have been unable to rescue hypersecretion observed in CpxI null mutants (*Wragg et al., 2017*). Moreover, single amino acid substitutions within the amphipathic helices of mouse and worm Cpx CTDs (*Kaeser-Woo et al., 2012*; *Wragg et al., 2013*), designed to interfere with vesicular membrane binding, may also compromise hydrophobic interactions of the CTDs with the SNARE proteins and curb the clamping ability of the mutants. Consistent with our hypothesis, recent work by Dittman and colleagues suggested that mutations within the amphipathic region of worm Cpx exhibited lipid binding in vitro but failed to rescue protein function in vivo (*Snead et al., 2017*; *Wragg et al., 2017*), indicating that generic membrane binding does not suffice for Cpx's inhibitory function. By adapting viral driven protein expression to levels seen in wildtype cells, we show that the CpxII:SN1 chimera indeed suppresses tonic secretion more efficiently than the full-length CpxII. In the same line, this mutant variant selectively prolongs the time-course of fusion pore expansion. Thus, structural similarity to SNAP25-SN1 seems to explain the enhanced clamp action of the mutant protein during pre- and postfusional stages. Notably, CpxII:SN1 chimera also reduces the spike-like current fluctuations of the early fusion pore – a finding that is remarkably similar to that of SybII linker mutants designed to interfere with force transfer on the merging membrane (*Kesavan et al., 2007*). These phenotypical similarities are in line with the view that the CpxII:SN1 chimera directly stalls the SNARE machinery. In contrast to the C-terminal half of the SNAP25-SN1 motif, the C-terminus of vertebrate CpxI and CpxII contains a Pro-Gly-Pro motif (PGP, see *Figure 4D*), which is predicted to interrupt the helicity of the

amphipathic region and may very well serve as a structural wedge to limit the clamping activity of CpxII in comparison to the CpxII:SN1 chimera.

Given that CpxII CTD interferes with SNARE zippering, one might expect a retardation of synchronous exocytosis. Yet, at the calcium concentrations tested here (15–25 µM), Ca$^{2+}$-binding to SytI has been shown to be rate-limiting for the stimulus-secretion coupling of chromaffin cells (*Voets et al., 2001*). Therefore, delayed and slower exocytosis timing in presence of the CpxII-CTD may become apparent only at very high [Ca]i (>80 µM, [*Sørensen et al., 2003*]). Since SNAREs act downstream of Ca$^{2+}$-triggering, it is also possible that SytI starts to lift the CpxII clamp at the moment of the Ca$^{2+}$-rise, thereby preventing retardation in exocytosis timing. In the same line, one might ask why infusion of the CTD peptide into wild type cells has a dominant effect even in the presence of the endogenous protein. An attractive explanation can be derived from single molecule FRET measurements, showing that Cys105 within the Cpx CTD produces only broad FRET efficiency peaks with the labeling site on the SNARE complex (Syntaxin 228, [*Bowen et al., 2005*]). This indicates conformational variability or motional averaging and suggests that even in the presence of endogenous CpxII the interaction of the CTD with SNAREs is transient and local concentrations of the CTD may not be saturating. Therefore, infusion of the CTD peptide or expression of full-length protein is able to enhance the inhibitory function of CpxII.

An intriguing facet of the proposed mechanism is added by the observation that the CpxII C-terminus fails to clamp tonic secretion in CpxII ko cells. Thus, other domains of the Cpx protein are required either to cooperate directly with the CTD or to establish a prefusion intermediate on which the C-terminus of CpxII can efficiently exert its clamp function (*Figure 8*). In any case, the most probable route to constrain the SNARE machinery before the triggering Ca$^{2+}$-stimulus will require the orchestration of multiple peptide domains from different proteins like Cpx and Syt in order to install a reliable clamp of the fusion apparatus.

Taken together, our experiments have pinpointed a so far unrecognized mechanism for Cpx's clamp action in Ca$^{2+}$-triggered exocytosis. They illustrate that structural similarities of Cpx's CTD with the SNAP25-SN1 domain are instrumental in clamping SNARE-mediated premature tonic secretion, leading to an increase in synchronous exocytosis. We propose that membrane binding and SNARE interactions of the CpxII CTD represent two independent modes of action that exist in a dynamic equilibrium and mediate on one hand the accumulation of Cpx at the site of vesicle fusion and on the other hand its inhibitory action on the fusion apparatus.

# Materials and methods

## Key resources table

| Reagent type (species) or resource | Designation | Source or reference | Identifiers | Additional information |
|---|---|---|---|---|
| Strain, strain background (*Mus musculus*) | C57BL/6 | | | |
| Genetic reagent (*Mus musculus*) | CPX II null allele | Reim et al., 2001. Complexins regulate a late step in Ca$^{2+}$-dependent neurotransmitter release. Cell, 104, 71–81 | PMID 11163241 | |
| Antibody | mouse anti-Syntaxin1 | Synaptic Systems | Cat# 110 001 | Western blot: 1:1000 |
| Antibody | mouse anti- SNAP25 | Synaptic Systems | Cat# 111 011 | Western blot: 1:1000 |
| Antibody | mouse anti-SynaptobrevinII | Synaptic Systems | Cat# 104 211 | ICC, Western blot: 1:1000 |
| Antibody | mouse anti-Synaptotagmin I | Synaptic Systems | Cat# 105 011 | Western blot: 1:1000 |
| Antibody | mouse anti-Synapsin 1 | Synaptic Systems | Cat# 106 001 | Western blot: 1:1000 |
| Antibody | rabbit anti-CpxII | this paper | Material and methods | ICC, Western blot 1:5000 |
| Antibody | HRP conjugated goat-anti mouse | Bio-Rad laboratories | Cat# 170–5047 | Western blot: 1:1000 |

*Continued on next page*

*Continued*

| Reagent type (species) or resource | Designation | Source or reference | Identifiers | Additional information |
|---|---|---|---|---|
| Antibody | HRP conjugated goat-anti rabbit | Bio-Rad laboratories | Cat# 170–5046 | Western blot: 1:1000 |
| Antibody | Alexa Fluor 555 goat anti-mouse | Invitrogen | Cat# A21422 | ICC: 1:1000 |
| Antibody | Alexa Fluor 488 goat anti-rabbit | Invitrogen | Cat# A11008 | ICC: 1:1000 |
| cDNA (*Mus musculus*) | CpxII-WT | GenBank: U35101.1 | | |
| Transfected construct (*Mus musculus*) | pSFV-Cpx1-100-IRES-EGFP | this paper | | derived from U35101.1 with indicated mutations |
| Transfected construct (*Mus musculus*) | pSFV-Cpx1-115-IRES-EGFP | this paper | | derived from U35101.1 with indicated mutations |
| Transfected construct (*Mus musculus*) | pSFV-Cpx1-100LC-IRES-EGFP | this paper | | derived from U35101.1 with indicated mutations |
| Transfected construct (*Mus musculus*) | pSFV-Cpx1-115SC-IRES-EGFP | this paper | | derived from U35101.1 with indicated mutations |
| Transfected construct (*Mus musculus*) | pSFV-Cpx1-100helix-IRES-EGFP | this paper | | derived from U35101.1 with indicated mutations |
| Transfected construct (*Mus musculus*) | pSFV-Cpx1-115helix-IRES-EGFP | this paper | | derived from U35101.1 with indicated mutations |
| Transfected construct (*Mus musculus*) | pSFV-Cpx1-100CSPα-IRES-EGFP | this paper | | derived from U35101.1 with indicated mutations |
| Recombinant DNA reagent | pGEX-KG-vector | this paper | | |
| Recombinant DNA reagent | pET-28a | this paper | | |
| Peptide, recombinant protein | C-terminal domain peptide | this paper | | Material and methods |
| Peptide, recombinant protein | scrambled peptide | this paper | | Material and methods |
| Software algorithm | IgorPro | WaveMetrics Software | | |
| Software algorithm | AutesP | Npi electronics | | |
| Software algorithm | Zen2008 | Zeiss | | |

## Mutagenesis and viral constructs

Substitution or truncation mutations in CpxII were generated by overlap extension polymerase chain reaction (PCR) using appropriate primers containing the desired non-homologous sequences (*Higuchi et al., 1988*). All mutations were confirmed by DNA sequence analysis (MWG Biotech, Germany). For expression in chromaffin cells, cDNAs encoding for CpxII or its mutant variants were subcloned into bicistronic Semliki Forest vector (pSFV1, Invitrogen, San Diego, CA). The vector contains an internal ribosomal entry site (IRES) controlled second open reading frame that encodes for enhanced green fluorescent protein (EGFP). This strategy allowed us to identify infected cell with EGFP expression (excitation wavelength 477 nm). Virus cDNA was linearized with restriction enzyme SpeI and transcribed in vitro by using SP6 RNA polymerase (Ambion, USA). BHK21 cells were transfected by electroporation (400V, 975 µF) with a combination of 10 µg CpxII (wildtype/mutant) and

pSFV-helper2 RNA. After 15 hr incubation (31°C, 5% $CO_2$), virus particles released into the supernatant were collected by low speed centrifugation (200 g, 5 min), snap-frozen and stored at −80°C (*Ashery et al., 1999*).

## Culture of chromaffin cells and electrophysiological recordings

All experiments were performed on mouse chromaffin cells prepared at postnatal day 0–1 from Complexin II knock out (CpxII$^{-/-}$) or littermate control (CpxII$^{+/+}$ or CpxII$^{+/-}$) animals (*Dhara et al., 2014*). Preparation of adrenal chromaffin cells was performed as described previously (*Borisovska et al., 2005*). Electrophysiological recordings were done on cultured chromaffin cells on the second or third day in culture and 5.5–6 hr after infection of cells with virus particles. Chromaffin granule secretion was stimulated by brief UV-flash that led to $Ca^{2+}$-uncaging upon photolysis of nitrophenyl-EGTA. Recordings of membrane capacitance (reflecting vesicle fusion) and ratiometric $[Ca^{2+}]_i$ changes (using Fura-2 and Furaptra) were performed as described previously (*Borisovska et al., 2005*). The intracellular solution for $Ca^{2+}$-uncaging experiments contained (in mM): 110 Cs-glutamate, 8 NaCl, 3.5 CaCl$_2$, 5 NP-EGTA, 0.2 fura-2, 0.3 furaptra, 2 MgATP, 0.3 Na$_2$GTP, 40 HEPES-CsOH, pH 7.3, 300 mOsm. The flash-evoked capacitance response was approximated with the function: $f(x)=A0+A1(1-\exp[-t/\tau1])+A2(1-\exp[-t/\tau2])+kt$, where A0 represents the cell capacitance before the flash. The parameters A1, τ1, and A2, τ2, represent the amplitudes and time constants of the RRP and the SRP, respectively (*Rettig and Neher, 2002*). The secretory delay was defined as the time between the UV-flash and the intersection point of the back-extrapolated fast exponential of capacitance rise with the baseline.

For simultaneous recordings of membrane capacitance and carbon fiber amperometry exocytosis was stimulated by infusion of an intracellular solution containing 19 µM free $Ca^{2+}$. The intracellular solution contained (in mM): 110 Cs-glutamate, 8 NaCl, 20 diethylene triamine penta-acetic acid, 5 CaCl$_2$, 2 MgATP, 0.3 Na$_2$GTP, and 40 Hepes-CsOH, pH 7.3 (osmolarity adjusted to 300 mOsm). The extracellular Ringer's solution used for all electrophysiological recordings contained (in mM): 130 NaCl, 4 KCl, 2 CaCl$_2$, 1 MgCl$_2$, 30 glucose, 10 HEPES-NaOH, pH 7.3 (osmolarity adjusted to 310 mOsm). Single amperometric spikes were recorded using home-made carbon fiber electrodes (ø 5 µm, Amoco), as described in (*Bruns, 2004*). Current signals were filtered at 2 kHz and digitized gap-free at 25 kHz prior to analysis. Only amperometric spikes with a peak amplitude > 4 pA and within the charge range from 10 to 5000 fC were considered for frequency analysis. Events with a peak amplitude > 7 pA were selected for the analysis of kinetic properties using Autesp (npi electronics, Tamm, Germany). To obtain reliable data for the current fluctuations and rms noise during prespike signal, we restricted our analysis to prespike duration longer than 2 ms. For the analysis of prespike signal flickers, the current derivative was further filtered at 1.2 kHz. Only deflections exceeding a threshold level of ± 6 pA/ms (corresponding to 4*SD) were considered and the fluctuation frequency was calculated as the number of suprathreshold fluctuations divided by the prespike duration. For the peptide infusion experiments, the intracellular solution contained, either the CpxII CTD-peptide (IPAGCGDEEEEEEESILDTVLKYLPGPLQDMFKK) or its scrambled variant (KVPYELGGQLPELKTSDPIE-GEDEDELFMKEIAC) at a final concentration of 10 µM.

## Biochemistry

### Pull-down assay

SulfoLink Coupling Gel (Pierce) was used for immobilization of the CpxII CTD peptide and its scrambled variant with an additional C-terminal cysteine according to the manufacturer's instructions (1.0 mg peptides/ml gel). Triton X-100 extract of mouse brain homogenate (0.5 mg/ml, containing 130 mM NaCl, 50 mM HEPES-NaOH, 1 mM EDTA, 2% Triton X-100, 1 mM PMSF, pH 7.3) was applied to beads (250 µl) and bound proteins were analyzed by 12% SDS-PAGE and Western blotting. For the detection of Syntaxin 1a, SNAP25, SybII, SytI and Synapsin1 the following mouse antibodies from Synaptic systems (Göttingen, Germany) were used: Anti-Syntaxin1 (CL 78.2) No: 110 001; SNAP25 (CL 71.1), No:111 011; SynaptobrevinII (CL 69.1), No: 104 211; Synaptotagmin 1 (CL 41.1), No:105 011; Anti-Synapsin 1 (CL 46.1), No:106 001. Primary mouse antibodies were used at a dilution of 1:1000. Immunoreactive bands were visualized with secondary goat-anti mouse or goat anti-rabbit antibodies conjugated with horseradish peroxidase and with an enhanced chemiluminescence system (Thermo Fisher Scientific, Schwerte, Germany).

## Ternary SNARE complex assembly assay

Recombinant SNAP25 (amino acids 1–206) and Syx 1a (amino acids 1–262) were expressed with an N-terminal His$_6$ tag in *E. coli* (BL21DE3) and purified using nickel-nitrilotriacetic acid-agarose (Qiagen, Germany). Recombinant SybII (amino acids 1–116) was expressed as N-terminal tagged GST fusion protein (pGEX-KG-vector) in the *E. coli* strain BL21DE3 and purified using glutathione-agarose according to the manufacturer's instructions. All column elutes were analyzed for integrity and purity of the expressed proteins by SDS-PAGE and staining with Coomassie blue. Binary t-SNARE complexes were preformed for 1 hr by mixing SNAP25 and Syx 1a at 5:1 molar ratio to facilitate 1:1 acceptor complex formation (*Pobbati et al., 2006*). The CpxII CTD peptide or its scrambled variant (50 µM) was incubated with the binary complex for 30 min before SybII (3 µM) was added to start complex assembly. The binding buffer contained (in mM): 100 NaCl, 1 DTT, 1 EDTA, 0.5% Triton X-100, 20 Tris (pH 7.4). Ternary SNARE complex formation was assayed at the indicated time points (*Figure 4*) and assembly reactions were stopped by adding 5xSDS sample buffer. The formation of SDS-resistant complexes was analyzed by SDS-PAGE (without boiling the samples) and Coomassie blue staining of protein bands. The rate of complex formation was calculated by fitting the data (using Sigmaplot 12) with the equation $SC(t) = SC_0 + (SC_\infty - SC_0) (A_{0*}k*t)/A_{0*}k*t + 1$ (*Nicholson et al., 1998*; *Pobbati et al., 2006*). This equation is derived from the second order reaction $A + B -> P$, where $A_0 = B_0$. SC(t) is the integrated density value of assembled SNARE complexes at time t, $SC_0$ is the experimental value at t = 0, $SC_\infty$ is the experimental value at t=∞. $A_0$ is the initial reactant concentration (M) and k is the rate constant ($M^{-1} S^{-1}$). The kinetic rates of SNARE complex formation in different groups (no peptide addition vs addition of CTD-peptide or scr-peptide) were tested for statistically significant difference using ANOVA on ranks with Dunn's post test.

## Analysis of structural similarity

Similarity scores between the CpxII CTD and the SNARE proteins were calculated using the BLOSUM62 matrix (EMBOSS needle). For the calculation of the hydrophobic moment the equation by (*Eisenberg et al., 1982*) and EMBOSS hmoment tool were used.

## Immuncytochemistry

For immunolabeling, chromaffin cells were processed either 5.5 hr (*Figure 6—figure supplement 2*) or 3.5 hr (*Figure 7A*) after virus infection as described previously (*Borisovska et al., 2012*). Epifluorescence images (eight bit encoded) were acquired with an AxioCam MRm-CCD camera (Carl Zeiss, Inc.) and analyzed with ImageJ software version 1.45. A homemade, affinity purified rabbit polyclonal antibody against CpxII (epitope: amino acids 1–100 of CpxII) was used for all immunofluorescence experiments described in the manuscript. The total intensity of the fluorescent immunolabel was quantified within the cytoplasm of the cell, which was determined by subtracting the nuclear fluorescence from total cellular fluorescence (area of interest comprising the outer cell perimeter – area of interest comprising the cell nucleus). In colocalization experiments, chromaffin cells were co-stained with rabbit polyclonal CpxII and mouse monoclonal SybII antibodies (clone 69.1, antigen epitope amino acid position 1–14, kindly provided by R. Jahn, MPI for Biophysical Chemistry, Göttingen, Germany). Images (16 bit encoded) were acquired at the 'foot-print' area of the cells with confocal microscope (LSM 710; Carl Zeiss) using excitation light of 488 and 555 nm wavelengths and the AxioVision 2010 software (Carl Zeiss) through a 100x, 1.3 NA oil objective. Immunopositive signals were determined after threshold adjustment (4x background signal) and cytofluorgram as well as Pearson's co-localization coefficient were analyzed with ImageJ (JACoP plugin).

## Statistics

Statistical tests were performed in Sigmaplot 12 (Systat Software). All data was tested for statistical significance with Student's t-test between two groups or one way analysis of variance (ANOVA) followed by Tukey-Kramer post-test for multiple conditions, if not indicated otherwise. Significance levels: '*' $p < 0.05$, '**' $p < 0.01$, and '***' $p < 0.001$.

## Acknowledgement

We thank Marina Wirth, Vanessa Schmitt for excellent technical assistance and Drs. J Rettig and D Stevens for helpful discussions. We would like to express our gratitude to C Fecher-Trost for performing the mass spectrometric analysis of the oligopeptides. This work was supported by the Collaborative Research Center 894 to DB and MJ.

## Additional information

### Funding

| Funder | Grant reference number | Author |
|---|---|---|
| Deutsche Forschungsgemeinschaft | SFB894 | Martin Jung Dieter Bruns |

The funders had no role in study design, data collection and interpretation, or the decision to submit the work for publication.

### Author contributions

Mazen Makke, Maria Mantero Martinez, Surya Gaya, Data curation, Formal analysis; Yvonne Schwarz, Resources, Data curation; Walentina Frisch, Lina Silva-Bermudez, Data curation; Martin Jung, Resources; Ralf Mohrmann, Writing—review and editing; Madhurima Dhara, Conceptualization, Data curation, Formal analysis, Writing—original draft, Project administration, Writing—review and editing; Dieter Bruns, Conceptualization, Formal analysis, Funding acquisition, Writing—original draft, Project administration, Writing—review and editing

### Author ORCIDs

Surya Gaya http://orcid.org/0000-0003-0163-5748
Martin Jung http://orcid.org/0000-0002-1482-7020
Madhurima Dhara http://orcid.org/0000-0001-7745-472X
Dieter Bruns http://orcid.org/0000-0002-2497-1878

### Decision letter and Author response

Decision letter https://doi.org/10.7554/eLife.38981.021
Author response https://doi.org/10.7554/eLife.38981.022

## Additional files

### Supplementary files

• Transparent reporting form
DOI: https://doi.org/10.7554/eLife.38981.019

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
