## [Decision Letter]

Thank you for submitting your article "A mechanism for exocytotic arrest by the Complexin C-terminus" for consideration by *eLife*. Your article has been reviewed by three peer reviewers, one of whom is a member of our Board of Reviewing Editors, and the evaluation has been overseen by Eve Marder as the Senior Editor. The reviewers have opted to remain anonymous.

The reviewers have discussed the reviews with one another and the Reviewing Editor has drafted this decision to help you prepare a revised submission.

Summary:

Using calcium-dependent exocytosis in cultured mouse chromaffin cells as model, Makke et al., identify a novel interaction between the C-terminal region of complexin II and the neuronal SNARE-complex. This binding stabilizes pools of primed granules, prevents tonic exocytosis and prolongs the duration of the fusion pore. Further analysis leads to the suggestion that the C-terminal does this by interfering with the formation of SNARE-complexes due to structural similarities with the SNAP25 SN1-domain, resulting in a competition for binding. The conclusions are based on capacitance measurements after expression of mutants or complexin:SNAP25 chimeras, infusion of peptides, single-spike amperometry, in vitro SNARE-complex formation immunostainings, and quantitative measurements of SNARE complexes using SDS-PAGE.

All reviewers agree that this is a highly interesting study that is of exceptional quality and presented in a scholarly manner.

Reviewer 1:

1) The biochemical support for the mechanistic explanation is circumstantial but not conclusive. The retardation of assembly (Figure 4C) is not overly convincing. Moreover, the authors should at least discuss that the putative coiled-coil motif of CpxII contains two helix-breaking proline residues in the middle of the stretch, making it hard to envision how this segment can form an α-helix. To strengthen the interpretation, direct binding studies would be required using purified proteins, preferably in comparison (or in competition) with the corresponding SN1 fragment. I realize that such experiments would require considerable additional work, which goes beyond the scope of the study.

2) If SNARE-zippering is impaired, then one would expect a reduction (or at least retardation) of synchronous release, particularly under conditions of sub-maximal stimulation. However, it appears that this is not the case. This (and many other models discussed at present) ignore completely that any interference with zippering requires additional energy to be overcome during triggering, thus reducing the energetic yield of the final zippering reaction. This should be at least discussed.

3) The model also has difficulties in explaining the puzzling observation that infusion of the peptide in a wt background has a "dominant" phenotype. It is highly unlikely that a fragment, added in solution, has an advantage in binding in comparison to the locally bound wt protein that is already positioned for binding, with a much higher apparent concentration.

Thus, alternative scenarios can be envisioned. For instance, it may be possible that the peptide, rather than blocking SNARE zippering, inactivates some SNARE complexes. It is frequently overlooked that more loosely interacting SNARE complexes may form a bit farther away from the membrane contact site. Thus, a core of correctly positioned SNARE complexes may be surrounded by less well-organized complexes of different stability and different degree of association with each other or with accessory proteins. Such complexes may be less tightly controlled by synaptotagmin and may result in accidental firing and occasional fusion. Steric requirements for blocking or reducing the activity of such loose, less ordered and perhaps dynamic trans-complexes may be lower, explaining why peptides with a less than optimal fit are capable of inhibition.

Reviewer #2:

1) Figure 1D: was the increase in RRP size by Cpx^1-100^ not significant? That would be expected given the error bars, but is not indicated. This figure and otherwise: what does it mean when the authors state: ANOVA between indicated groups. An ANOVA does not allow to determine which groups are different from each other. According to the Materials and methods, a Tukey-Kramer test was used as post-hoc test. Please confirm in each figure legend that the levels of significance are those resulting from the TK-test.

2) In peptide infusion experiments: how long was the wait after going whole-cell, until the peptide affected secretion properties, for instance the foot duration? One would expect that this could not be instantaneous, but would have to develop over time, as the peptide infused through the narrow pipette tip. Yet, in Figure 2D, 2E and 3E,G it appears that the effect is instantaneous. The authors should explain whether the time zero, in their infusion experiments, is the time of breaking through the patch to obtain the whole-cell configuration. If this is so, how do the authors explain that the peptide acts instantaneously? The authors should estimate the time constant for infusion given the molecular weight of the peptide (Pusch et al., 1988). If the effect is very rapid, the concentration in the pipette must be much higher than the needed concentration in the cell, and this is probably worth discussing.

3) A cartoon showing how the complexin C-terminal can block SNARE-complex assembly might help getting the point across.

Reviewer #3

The only issue I see is that the authors did not monitor resting spontaneous release in this preparation (as I gather from the methods, even tonic release is triggered by buffered Ca^2+^ infusion through the recording pipette). I think assessing the impact of the manipulations reported in this study on unstimulated true spontaneous release (in resting Ca^2+^) will provide a better basis for comparison to experiments conducted in synapses. Given the vast amount of data included in this manuscript, I believe the authors may already have this data on baseline release properties.

---

## [Author Response]

Reviewer 1:1) The biochemical support for the mechanistic explanation is circumstantial but not conclusive. The retardation of assembly (Figure 4C) is not overly convincing. Moreover, the authors should at least discuss that the putative coiled-coil motif of CpxII contains two helix-breaking proline residues in the middle of the stretch, making it hard to envision how this segment can form an α-helix.

We agree with the reviewer that helix-breaking proline residues may intercept the formation of a continuous α-helix and would like to highlight that the Discussion section of the original manuscript already pointed out this issue in subsection “A new model for the clamp function of CpxII C-terminus”:

“In contrast to the C-terminal half of the SNAP25-SN1 motif, the C-terminus of vertebrate CpxI and CpxII contains a Pro-Gly-Pro motif (PGP, see Figure 4D), which is predicted to interrupt the helicity of the amphipathic region and may very well serve as a structural wedge to limit the clamping activity of CpxII in comparison to the CpxII:SN1 chimera.”

For the information of the reviewer we show the protein structure prediction for the CpxII CTD (aa101-134):

Sequence (Tasser 3D protein prediction)

IPAGCGDEEEEEEESILDTVLKYLPGPLQDMFKK

CCCCCCCCCHHHHHHHHHHHHHHCCCHHHHHHCC Pred.

9998788701548889999999868469988519 Conf.Sc.

C:Coil, H:Helix, S:Strand

In good agreement with his/her suggestion and with our discussion the helical structure of the CpxII CTD is predicted to be interrupted by the short PGP motif. As pointed out above, such structural feature might be important for reversing the Cpx-CTD clamp on partially assembled complex allowing for timely completion of SNARE-zippering and vesicle fusion.

To strengthen the interpretation, direct binding studies would be required using purified proteins, preferably in comparison (or in competition) with the corresponding SN1 fragment. I realize that such experiments would require considerable additional work, which goes beyond the scope of the study.

We agree with the reviewer that protein binding assays between Cpx and SN1 motifs with purified proteins would be useful to strengthen and critically test the model how Cpx-CTD blocks SNARE complex assembly. We are currently establishing such binding assays using recombinant proteins. To study Cpx:SNARE interaction in a comprehensive fashion requires extensive preparatory work and considerable additional time. Thus, we very much appreciate the reviewer’s fair judgement that such detailed biochemical assays are beyond the scope of the present study.

2) If SNARE-zippering is impaired, then one would expect a reduction (or at least retardation) of synchronous release, particularly under conditions of sub-maximal stimulation. However, it appears that this is not the case. This (and many other models discussed at present) ignore completely that any interference with zippering requires additional energy to be overcome during triggering, thus reducing the energetic yield of the final zippering reaction. This should be at least discussed.

We are grateful for the reviewer’s comment and now discuss this issue in in subsection “A new model for the clamp function of CpxII C-terminus”:

“Given that CpxII CTD interferes with SNARE zippering, one might expect a retardation of synchronous exocytosis. Yet, at the calcium concentrations tested here (15-25 μM), Ca^2+^-binding to SytI has been shown to be rate-limiting for the stimulus-secretion coupling of chromaffin cells (Voets et al., 2001). Therefore, delayed and slower exocytosis timing in presence of the CpxII-CTD may become apparent only at very high [Ca]i (>80 µM, Sorensen et al., 2003). Since SNAREs act downstream of Ca^2+^-triggering, it is also possible that SytI starts to lift the CpxII clamp at the moment of the Ca^2+^-rise, thereby preventing retardation in exocytosis timing.”

3) The model also has difficulties in explaining the puzzling observation that infusion of the peptide in a wt background has a "dominant" phenotype. It is highly unlikely that a fragment, added in solution, has an advantage in binding in comparison to the locally bound wt protein that is already positioned for binding, with a much higher apparent concentration.Thus, alternative scenarios can be envisioned. For instance, it may be possible that the peptide, rather than blocking SNARE zippering, inactivates some SNARE complexes. It is frequently overlooked that more loosely interacting SNARE complexes may form a bit farther away from the membrane contact site. Thus, a core of correctly positioned SNARE complexes may be surrounded by less well-organized complexes of different stability and different degree of association with each other or with accessory proteins. Such complexes may be less tightly controlled by synaptotagmin and may result in accidental firing and occasional fusion. Steric requirements for blocking or reducing the activity of such loose, less ordered and perhaps dynamic trans-complexes may be lower, explaining why peptides with a less than optimal fit are capable of inhibition.

This point is also well taken. Accordingly, we have now discussed this issue in subsection “A new model for the clamp function of CpxII C-terminus” as follows:

“In the same line, one might ask why infusion of the CTD peptide into wild type cells has a dominant effect even in the presence of the endogenous protein. An attractive explanation can be derived from single molecule FRET measurements, showing that Cys105 within the Cpx CTD produces only broad FRET efficiency peaks with the labeling site on the SNARE complex (Syntaxin 228, Bowen et al., 2005). This indicates conformational variability or motional averaging and suggests that even in the presence of endogenous CpxII the interaction of the CTD with SNAREs is transient and local concentrations of the CTD may not be saturating. Therefore, infusion of the CTD peptide or expression of full-length protein is able to enhance the inhibitory function of CpxII.”

Reviewer #2:1) Figure 1D: was the increase in RRP size by Cpx^1-100^ not significant? That would be expected given the error bars, but is not indicated. This figure and otherwise: what does it mean when the authors state: ANOVA between indicated groups. An ANOVA does not allow to determine which groups are different from each other. According to the Materials and methods, a Tukey-Kramer test was used as post-hoc test. Please confirm in each figure legend that the levels of significance are those resulting from the TK-test.

We carefully checked the statistical analyses between CpxII ko, CpxII ko+CpxII and CpxII ko+ Cpx^1-100^ and found no significant difference between the RRP values of CpxII ko and CpxII ko+ Cpx^1-100^ (p=0.173, ANOVA followed by Tukey-Kramer post-hoc test between the groups).

As suggested by the reviewer, we have mentioned now the use of Tukey-Kramer post-test for determination of significance values in the respective figure legends.

2) In peptide infusion experiments: how long was the wait after going whole-cell, until the peptide affected secretion properties, for instance the foot duration? One would expect that this could not be instantaneous, but would have to develop over time, as the peptide infused through the narrow pipette tip. Yet, in Figure 2D, 2E and 3E,G it appears that the effect is instantaneous. The authors should explain whether the time zero, in their infusion experiments, is the time of breaking through the patch to obtain the whole-cell configuration. If this is so, how do the authors explain that the peptide acts instantaneously? The authors should estimate the time constant for infusion given the molecular weight of the peptide (Pusch et al., 1988). If the effect is very rapid, the concentration in the pipette must be much higher than the needed concentration in the cell, and this is probably worth discussing.

We agree with the reviewer, that this is an important point and have added the following paragraph in the Results section of the revised:

“Of note, in a subset of recordings the initial CM increase of CTD peptide infused cells was similar to that of controls, before it rapidly declined to a slower, sustained increase in CM (Figure 2E, inset). Based on an estimated time constant of around 60 s for peptide infusion into chromaffin cells (R_access_ 11.8 ± 0.2 MΩ; CM 4.0 ± 0.1 pF; (Pusch and Neher, 1988)), this behavior suggests that saturating concentrations for inhibitory peptide action are readily reached after establishment of whole-cell configuration. Since in vitro single vesicle-vesicle fusion assays have shown that Cpx suppresses spontaneous fusion at concentrations as low as 0.5 µM (Lai et al., 2014), it is likely that the pipette concentration of CTD-peptide (10 µM) used in our experiments exceeds the required effective concentration of CpxII-CTD for inhibiting vesicle fusion.”

Accordingly, the corresponding legend of Figure 2E has been changed:

“(E) The CTD-pep reduces tonic secretion (upon infusion with 19 µM [Ca^2+^]i) in wt cells, but fails to do so in CpxII ko cells (t=0 is the time point about 10-15 sec after establishing the whole-cell configuration). The inset displays the initial CM response of wt cells infused with scr-pep (black) or CTD-pep (blue) at the expanded time scale.”

3) A cartoon showing how the complexin C-terminal can block SNARE-complex assembly might help getting the point across.

As suggested by the reviewer, we have included a cartoon depicting how CpxII C-terminus may interact with partially assembled SNARE complex and prevent the completion of SNARE zippering (Figure 8).

Reviewer #3The only issue I see is that the authors did not monitor resting spontaneous release in this preparation (as I gather from the methods, even tonic release is triggered by buffered Ca^2+^ infusion through the recording pipette). I think assessing the impact of the manipulations reported in this study on unstimulated true spontaneous release (in resting Ca^2+^) will provide a better basis for comparison to experiments conducted in synapses. Given the vast amount of data included in this manuscript, I believe the authors may already have this data on baseline release properties.

In our previous publication (Dhara et al., 2014, Figure 5 C-E), we have systematically investigated the effects of CpxII loss at different [Ca]i. We found that at intracellular Ca^2+^ concentrations of ~100 nM (near the resting cellular concentration, Sabatini and Regehr, 1998) the rate of granule exocytosis is nearly identical in wt (0.7 ± 0.8 fF/10s) and CpxII ko (1.1 ± 1 fF/10s) cells. Only at elevated [Ca^2+^]i, which is conducive for vesicle mobilization and ‘priming’ (≥350 nM), loss of CpxII leads to significantly higher release in chromaffin cells. Since chromaffin cells lack true neuronal ‘active zone’ like structures (Pang and Sudhof, 2010), it is likely that they require elevated [Ca^2+^]i for establishing the membrane proximal ‘primed vesicle pool’.

Thus inhibitory effects of CpxII are unmasked only at these elevated [Ca^2+^]i levels.